# A Global Markov Property for Solutions of Stochastic Difference Equations and the corresponding Full Time Graphs

**Tom Hochsprung**[1]        **Jakob Runge**[*1,2]        **Andreas Gerhardus**[*1]

[1]German Aerospace Center (DLR), Institute of Data Science, Jena, Germany
[2]Technische Universität Berlin, Berlin, Germany

## Abstract

Structural Causal Models (SCMs) are an important tool in causal inference. They induce a graph and if the graph is acyclic, a unique observational distribution. A standard result states that in this acyclic case, the induced observational distribution satisfies a $d$-separation global Markov property relative to the induced graph. Time series can also be modelled like SCMs: One just interprets the stochastic difference equations that a time series solves as structural equations. However, technical problems arise when time series "start" at minus infinity. In particular, a $d$-separation global Markov property for time series and the corresponding infinite graphs, the so-called full time graphs, has thus far only been shown for stable vector autoregressive processes with independent finite-second-moment noise. In this paper, we prove a much more general version of this Markov property. We discuss our assumptions and study violations of them. Doing so hints at several pitfalls at the intersection of time series analysis and causal inference. Moreover, we introduce a new projection procedure for these infinite graphs which might be of independent interest.

## 1 INTRODUCTION

Structural Causal Models (SCMs), also known as Structural Equation Models (SEMs), are widely used in causal inference [Pearl, 2009, Peters et al., 2017, Bongers et al., 2021]. An SCM consists of finitely many exo- and endogeneous random variables, a joint probability distribution over the exogeneous random variables, and assignment functions that causally relate the random variables to each other. SCMs induce a graph where each variable is represented by a node and each direct causal relation by a directed edge. When

---
[*]Equal supervision

the induced graph is acyclic, an SCM induces a unique observational distribution [Bongers et al., 2021, Proposition 3.4]. A standard result states that in this acyclic case, the induced observational distribution satisfies a $d$-separation global Markov property relative to the induced graph (Theorem 1.4.1 in Pearl [2009] combined with Proposition 4 in Lauritzen et al. [1990]). This result, also known as the causal Markov property, allows one to read off conditional independencies from the induced graph and hence is highly important for SCM-based causal reasoning.

SCMs frequently rely on independent and identically distributed (i.i.d.) observations. However, many real-world observations are not i.i.d. and instead come as time series. Nevertheless, one can still model many time series like SCMs: One just interprets the stochastic difference equations that time series often solve as structural equations. This approach runs under various names and assumptions: Structural vector autoregressive (SVAR) processes [Demiralp and Hoover, 2003, Hyvärinen et al., 2010, Moneta et al., 2011, Malinsky and Spirtes, 2018, Pamfil et al., 2020], additive nonlinear time series causal models [Chu et al., 2008], time series models with independent noise [Peters et al., 2013, 2017], vector autoregressive (VAR) processes [Dahlhaus and Eichler, 2003, Eichler, 2010, Entner and Hoyer, 2010, Thams et al., 2022], structural causal processes Runge [2020] or just SCMs [Gerhardus and Runge, 2020].

When the time points $t$ are in some finite index set $I$, no new technical difficulties arise and one can directly translate the stochastic difference equations into structural equations. Similarly in the case $t \in \mathbb{N}_{\geq 0}$: Even though the stochastic difference equations continue infinitely to the future, the fact that one can push-forward a given distribution over the noise variables still allows for a typical SCM-like interpretation. Technically more problematic is the case $t \in \mathbb{Z}$: Here, the stochastic process "starts" at $-\infty$. Therefore, it is not immediately clear what the induced distribution of a "time-series SCM" is. In fact, the stochastic difference equations might have no solution, one solution or several solutions. Moreover, these solutions might behave quite differently

and carry quite different causal interpretations: For example, solutions might depend on future noise variables, which does not fit well to the idea that causes precede their effect.

One might be tempted to just ignore the case $t \in \mathbb{Z}$. However, several reasons speak against doing so (especially when modelling equilibriums): First, thinking about stochastic difference equations that continue infinitely to the past is often more convenient than to additionally also think about initial distributions. Second, the notion of stationarity, on which most of the existing time series literature relies [Fan and Yao, 2003, Lütkepohl, 2005, Brockwell and Davis, 2009, among others], fits much more naturally to $t \in \mathbb{Z}$: When time series do not "start" at $-\infty$, then they usually only converge to a stationary state instead of always being in it (unless one particularly crafts initial distributions).

In this paper, we focus on one particular issue for $t \in \mathbb{Z}$: In spirit of Theorem 1.4.1 in Pearl [2009], we show that certain solutions of stochastic difference equations satisfy a $d$-separation global Markov property with respect to the induced infinite graph when $t \in \mathbb{Z}$. To the best of our knowledge, such a result has so far only been shown for stable vector autoregressive (VAR) processes with independent finite-second-moment noise [Thams et al., 2022].[1] We also discuss the required assumptions and study violations of them. Doing so hints at several pitfalls at the intersection of time series analysis and causal inference. Moreover, we thus also provide a better theoretical foundation for much existing and future work.

We structure our paper as follows: In **Section 2**, we introduce our notation and setup. **Section 3** is about proving the global Markov property. In **Section 4**, we show that our result applies to stable VAR processes with independent finite-second-moment noise that has a density with respect to Lebesgue measure. In **Section 5**, we provide a conclusion and an outlook. The **Supplementary Material (SM)** contains several of our proofs and further lemmas.

We require that the reader is familiar with basic terminology from causal inference. In particular, we use concepts such as graphs, $d$- and $m$-separation, conditional independence and SCMs. For a brief overview of graphs, $d$- and $m$-separation, see Section A in the SM, and for further reference see Peters et al. [2017], Pearl [2009] and Richardson [2003].

---

[1]Such a result also appears in Dahlhaus and Eichler [2003]. However, as also remarked by an anonymous reviewer, Dahlhaus and Eichler [2003] apply a result for finite graphs, namely the AMP Markov property from Andersson et al. [2001], to infinite graphs without giving further justification on why this is possible.

Besides, there are also several Markov properties for finite graphical representations of time series, see, e.g., Eichler [2012].

## 2 SETUP

In **Section 2.1**, we introduce notation that is relevant to the main paper (Section B.1 in the SM contains further notation that is just relevant to the SM). In **Section 2.2**, we connect causality and stochastic difference equations and introduce several concepts along the way. In that section, we also formally state the $d$-separation global Markov property. Finally, **Section 2.3** focuses on further technical assumptions that we need in order to prove the $d$-separation global Markov property.

### 2.1 NOTATION

For two sets $A$ and $B$, we write $A \times B$ to denote their Cartesian product. We write $\mathbb{Z}$ to denote the integers. For some $d \in \{0, 1, \ldots\}$, we write $\mathbb{N}_{\geq d}$ to denote the set $\{d, d+1, \ldots\}$. For some $k \in \mathbb{N}_{\geq 1}$ and some set $A$, we write $A^k$ to denote the $k$-times Cartesian product $A \times \ldots \times A$. For some $k \in \mathbb{N}_{\geq 1}$, we write $[k]_1$ to abbreviate the set $\{1, \ldots, k\}$. Similarly, for some $k \in \mathbb{N}_{\geq 0}$, we write $[k]_0$ to abbreviate the set $\{0, \ldots, k\}$. For some $k \in \mathbb{N}_{\geq 1}$ and sets $A_1, \ldots, A_k$, we write $\Pi_{i \in [k]_1} A_i$ to denote the set $A_1 \times \ldots \times A_k$, similarly for some $k \in \mathbb{N}_{\geq 0}$. To denote a vector with potentially more than one component (with respect to some base space), we use **bold font**.

For random vectors $\boldsymbol{X}$ and $\boldsymbol{Y}$ that are defined on the same underlying probability space $(\Omega, \mathcal{F}, P)$, we write $P_{\boldsymbol{X}\boldsymbol{Y}}$ to denote the joint distribution of $\boldsymbol{X}$ and $\boldsymbol{Y}$. For the marginal distributions of $\boldsymbol{X}$ and $\boldsymbol{Y}$, we write $P_{\boldsymbol{X}}$ and $P_{\boldsymbol{Y}}$, respectively. To denote unconditional independence between $\boldsymbol{X}$ and $\boldsymbol{Y}$, we write $\boldsymbol{X} \perp\!\!\!\perp \boldsymbol{Y}$. To denote conditional independence between $\boldsymbol{X}$ and $\boldsymbol{Y}$ given some further random vector $\boldsymbol{Z}$, we write $\boldsymbol{X} \perp\!\!\!\perp \boldsymbol{Y} \mid \boldsymbol{Z}$. In slight abuse of notation, we consider a set $\boldsymbol{A}$ of random vectors as a random vector whose components are the in some way ordered elements of $\boldsymbol{A}$. Correspondingly, we write these sets of random vectors in bold font as well.

For the graphical part, we follow the convention that a vertex $v$ is never a parent or a child of itself (unless there is a self-edge, which we do not consider in this paper). However, we follow the convention that a vertex $v$ is always an ancestor or a descendant of itself. Moreover, for two vertices $a$ and $b$, we write $a *\!\!-\!\!* b$ to represent $a \to b$, $a \leftarrow b$ and $a \leftrightarrow b$ all at once. For more details and more precise explanations, see Section A in the SM.

### 2.2 CAUSALITY AND STOCHASTIC DIFFERENCE EQUATIONS

Let $(\Omega, \mathcal{F}, P)$ be a probability space and let $d \in \mathbb{N}_{\geq 1}$. For all $i \in [d]$, let $(\mathcal{X}_i, \Sigma_i^X)$ and $(\mathcal{E}_i, \Sigma_i^\epsilon)$ be measurable spaces such that all $\mathcal{X}_i$ and all $\mathcal{E}_i$ are either real finite-dimensional vector spaces or finite discrete sets and such that $\Sigma_i^X$ and $\Sigma_i^\epsilon$

are the corresponding Borel $\sigma$-algebras. We consider a given exogenous noise process $\{\boldsymbol{\epsilon}_t\}_{t\in\mathbb{Z}} := \{(\epsilon_t^1, \ldots, \epsilon_t^d)\}_{t\in\mathbb{Z}}$ and a given endogenous process $\{\boldsymbol{X}_t\}_{t\in\mathbb{Z}} := \{(X_t^1, \ldots, X_t^d)\}_{t\in\mathbb{Z}}$. Here, each $X_t^i$ and each $\epsilon_t^i$ is a random variable defined on $(\Omega, \mathcal{F}, P)$ and taking values in $(\mathcal{X}_i, \Sigma_i^X)$ respectively $(\mathcal{E}_i, \Sigma_i^\epsilon)$. Similarly to existing work [Peters et al., 2013, Thams et al., 2022], we assume that

- **(A0)**: $\{\boldsymbol{\epsilon}_t\}_{t\in\mathbb{Z}}$ is an i.i.d. process with independent components, that is, the distribution of any finite subset of $\{\epsilon_t^i : i \in [d], \ t \in \mathbb{Z}\}$ factorizes into its individual components and each $\boldsymbol{\epsilon}_t$ has the same distribution.[2]

We further assume that $\{(\boldsymbol{X}_t, \boldsymbol{\epsilon}_t)\}_{t\in\mathbb{Z}}$ is a solution of given stochastic difference equations: For each $i \in [d]_1$, let $q_i \in \mathbb{N}_{\geq 0}$ be a fixed number (which is the order of the stochastic difference equation for $X_t^i$). Now, for each $i \in [d]_1$ and for each $s \in [q_i]_0$, let $\mathbf{PA}_s^i$ denote a tuple of fixed component time series, say the first and second component time series, and let $\mathcal{X}_{\mathrm{pa}_s^i}$ denote the corresponding product space. Furthermore, write $\mathcal{X}_{\mathrm{pa}^i} := \prod_{s\in[q_i]_0} \mathcal{X}_{\mathrm{pa}_s^i}$ and let $(\mathbf{PA}_s^i)_{t-s}$ denote the respective component time series evaluated at time point $t-s$.[3] Additionally, for all $i \in [d]_1$, let $f_i : \mathcal{X}_{\mathrm{pa}^i} \times \mathcal{E}_i \to \mathcal{X}_i$ be a measurable function. Now, for all $i \in [d]$ and $t \in \mathbb{Z}$, we assume that $\{(\boldsymbol{X}_t, \boldsymbol{\epsilon}_t)\}_{t\in\mathbb{Z}}$ solves $P$-almost surely the stochastic difference equation

$$X_t^i = f_i\left((\mathbf{PA}_{q_i}^i)_{t-q_i}, \ldots, (\mathbf{PA}_0^i)_t, \epsilon_t^i\right). \qquad (1)$$

**Remark 1.** *On the distribution level, a solution $\{(\boldsymbol{X}_t, \boldsymbol{\epsilon}_t)\}_{t\in\mathbb{Z}}$ or for simplicity $\{\boldsymbol{X}_t\}_{t\in\mathbb{Z}}$ corresponds to the collection of all joint and marginal distributions of the respective random vectors.*

**A causal interpretation:** So far, the stochastic difference equation in (1) is just "algebraic". One can give (1) a causal interpretation by saying that all non-superfluous variables on the right hand-side of the equation cause the variable on the left-hand side of the equation. In resemblance to SCMs, one can emphasize this structural character of (1) by replacing the "=" with a ":=". In the next definition, we formally introduce the notion of non-superfluous random variables which we from now on call *causal parents* [Bongers et al., 2021, c.f. Definition 2.6].

**Definition 1** (Causal parent). For $i, j \in [d]_1$ and $s, t \in \mathbb{Z}$, we call $X_s^j$ a (causal) parent of $X_t^i$ if and only if there does not exist a measurable function[4] $\tilde{f}_i : \mathcal{X}_{\mathrm{pa}^i \setminus X_s^j} \times \mathcal{E}_i \to \mathcal{X}_i$

such that for $P_{\epsilon_t^i}$-almost every $e \in \mathcal{E}_i$ and for all $\boldsymbol{x} \in \mathcal{X}_{\mathrm{pa}^i}$,

$$X_t^i := f_i(\boldsymbol{x}, e)$$
$$\iff X_t^i := \tilde{f}_i(\boldsymbol{x}_{\setminus X_s^j}, e).$$

We similarly define parentship of a noise variable $\epsilon_t^j$ on $X_t^i$.

One can graphically represent (1) using the *(augmented) full time graph*.

**Definition 2** ((Augmented) full time graph). The *full time graph* is an infinite graph with vertex set $\{X_t^i : t \in \mathbb{Z}, \ i \in [d]_1\}$.[5] The full time graph only has directed edges: There is a directed edge from $X_s^j$ to $X_t^i$ if and only if $X_s^j$ is a causal parent of $X_t^i$.

The *augmented full time graph* consists of further vertices $\{\epsilon_t^i : t \in \mathbb{Z}, \ i \in [d]_1\}$ and a directed edge from $\epsilon_t^i$ to $X_t^i$ if the former is a causal parent of the latter.

**Remark 2.** *In our setting, there never is a directed edge from $X_s^j$ to $X_t^i$ when $s > t$. Moreover, there never is a directed edge from $\epsilon_s^i$ to $X_t^i$ when $s \neq t$.*

**Remark 3.** *As long as Assumption (A1) below is satisfied, we allow contemporaneous edges in the full time graph, that is, edges between vertices at the same time point (the term $(\mathbf{PA}_0^i)_t$ in Equation (1) hints at this option).*

**Example 1.** Consider a stochastic process $\{\boldsymbol{\epsilon}_t\}_{t\in\mathbb{Z}} = \{\epsilon_t^1, \epsilon_t^2\}_{t\in\mathbb{Z}}$ satisfying (A0). In addition, consider a stochastic process $\{\boldsymbol{X}_t\}_{t\in\mathbb{Z}} = \{X_t^1, X_t^2\}_{t\in\mathbb{Z}}$ satisfying

$$X_t^1 := 0.4 \cdot X_{t-1}^1 + 0.2 \cdot X_{t-2}^2 + \epsilon_t^1 \quad \text{and}$$
$$X_t^2 := 0.3 \cdot X_{t-1}^2 + \epsilon_t^2. \qquad (2)$$

Figure 1 shows the full time graph and the augmented full time graph.

**Acyclicity assumption:** For the remainder of the paper, we assume that

- **(A1):** the full time graph does not contain directed cycles.

Note that directed cycles in the full time graph (and hence in the augmented full time graph) never occur when there are no contemporaneous edges.

**A global Markov property**: For SCMs whose induced graph is a DAG (and who by definition have finitely many variables), the induced observational distribution satisfies

---

[2]Note that the factorization assumption is equivalent to all finite subsets of $\{\epsilon_t^i : i \in [d], \ t \in \mathbb{Z}\}$ being mutually independent. Also note that Assumption (A0) *does not* state that two different components $\epsilon_t^i$ and $\epsilon_t^j$ need to have the same distribution.

[3]This notation is partly from Peters et al. [2013] and Chapter 10 of Peters et al. [2017].

[4]The notation $\mathcal{X}_{\mathrm{pa}^i \setminus X_s^j}$ means that the space corresponding to

---

$X_s^j$ is removed from $\mathcal{X}_{\mathrm{pa}^i}$. Similarly, for $\boldsymbol{x} \in \mathcal{X}_{\mathrm{pa}^i}$, writing $\boldsymbol{x}_{\setminus X_s^j}$ means that the element corresponding to $X_s^j$ is removed.

[5]For notational simplicity, we do not make a distinction between random variables and their representation by vertices in the (augmented) full time graph. For us, variables = vertices.

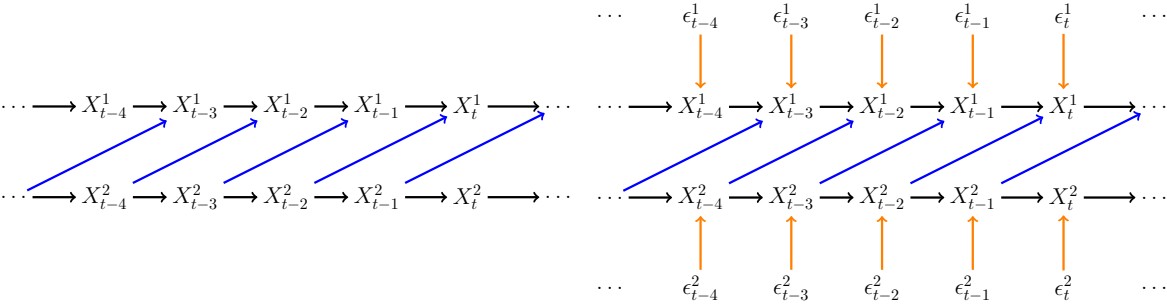

Figure 1: The full time graph (left) and the augmented full time graph (right) from Example 1. The "..." indicate that nodes and edges repeat infinitely to the past and future. We used color for better clarity.

a $d$-separation global Markov property with respect to the induced graph (Theorem 1.4.1 in Pearl [2009] combined with Proposition 4 in Lauritzen et al. [1990]). In this paper, we show that such a result also holds for the distributions of $\{\boldsymbol{X}_t\}_{t\in\mathbb{Z}}$ respectively $\{(\boldsymbol{X}_t, \boldsymbol{\epsilon}_t)\}_{t\in\mathbb{Z}}$ and the corresponding (augmented) full time graph. In the next theorem, we present this result; we defer the remaining technical assumptions to Section 2.3.

**Theorem 1** (A global Markov property for solutions of stochastic difference equations and the corresponding (augmented) full time graphs). *Every solution of the stochastic difference equations in* (1) *that satisfies (A0)–(A4) is globally Markov in a $d$-separation sense with respect to the full time graph. That is, for any finite disjoint index sets $J_1, J_2, J_3 \subseteq [d]_1 \times \mathbb{Z}$ such that $\boldsymbol{S}_3 := \{X_t^i : (i,t) \in J_3\}$ $d$-separates $\boldsymbol{S}_1 := \{X_t^i : (i,t) \in J_1\}$ and $\boldsymbol{S}_2 := \{X_t^i : (i,t) \in J_2\}$ in the full time graph, $\boldsymbol{S}_1 \perp\!\!\!\perp \boldsymbol{S}_2 \mid \boldsymbol{S}_3$.*

*An analogous result including noise variables $\{\epsilon_t^i : t \in \mathbb{Z}, i \in [d]_1\}$ holds for the augmented full time graph.*

### 2.3 FURTHER TECHNICAL ASSUMPTIONS

**Further technical assumptions**: In this paper, we further assume that

- **(A2):** $\{\boldsymbol{X}_t\}_{t\in\mathbb{Z}}$ is strictly stationary[6], that is, all distributions of finitely many vectors are time shift-invariant,

- **(A3):** $\{\boldsymbol{X}_s\}_{s\leq t}$ is independent of $\{\boldsymbol{\epsilon}_s\}_{s>t}$ for all $t \in \mathbb{Z}$ (meaning that all finite subsets of these two sets are independent), and

- **(A4):** $\{\boldsymbol{X}_t\}_{t\in\mathbb{Z}}$ is $\alpha$-strongly mixing, that is,

$$\alpha(m) := \sup_{A\in\mathcal{A}_0,\ B\in\mathcal{A}^m} |P(A\cap B) - P(A)P(B)|$$
$$\xrightarrow{m\to\infty} 0$$

where $\mathcal{A}_0 := \sigma(\boldsymbol{X}_t : t \leq 0)$ and $\mathcal{A}^m := \sigma(\boldsymbol{X}_t : t \geq m)$.[7]

**Discussion of the assumptions:** We decided to make assumptions that often appear in the time series (and causal inference) literature and from thereon prove the global Markov property.

Assumptions similar to (A2), so other forms of stationarity, are common in the time series literature [Brockwell and Davis, 2009, Fan and Yao, 2003, among others] and the time series causal inference literature [Malinsky and Spirtes, 2018, Pamfil et al., 2020, among others]. Furthermore, existing Markov properties for time series also assume some form of stationarity [Dahlhaus and Eichler, 2003, Eichler, 2012, Thams et al., 2022]. *Strict* stationarity in particular, so Assumption (A2), is the most natural stationarity-notion for the nonlinear setting as other weaker notions such as covariance stationarity neglect higher moments [Fan and Yao, 2003, Section 2.1.1]. The existing Markov property closest to our work from Thams et al. [2022] implicitly also assumes strict stationarity as we explain in Section 4.

Also note that nonstationarity is usually better modelled by allowing the functions in equation (1) to be time-variant instead of considering "artificial" nonstationary solutions to time-invariant equations. So while stationarity (in whatever form) is a strong assumption and many real-world time series arguably violate it, stationarity within the realm of our time-invariant difference equations looks rather natural. The strong assumption rather is equation (1); we consider it future work to generalize it.

Assumption (A3) is also common in the time series literature and as we argue in Section 4, implicitly made by Thams et al. [2022]. Assumption (A3) states that noise variables only influence present and future instances of $\{\boldsymbol{X}_t\}_{t\in\mathbb{Z}}$. Processes that do not satisfy assumption (A3) are known as "non-causal" or "future-dependent" [Brockwell and Davis, 2009, Section 3.1]. Note that a violation of (A3) also imme-

---

[6]The part of Assumption (A0) which states that each $\boldsymbol{\epsilon}_t$ has the same distribution ensures that Assumption (A2) can hold for non-pathological cases. However, this part of (A0) is not required for the proofs.

[7]Due to Assumption (A2), namely strict stationarity, we could take any other reference point than 0 (see, e.g., Bradley [2005] for this statement and Lemma 5 in the SM for a formal proof).

diately implies a violation of the global Markov property for the augmented full time graph. Besides, Assumption (A3) fits well to the existing causal inference literature: In SCMs, noise variables also only influence endogeneous variables further down the causal chain.

Lastly, Assumption (A4) is typical in the *nonlinear* time series literature and used to prove several limit theorems [Fan and Yao, 2003, Section 2.6]. Assumption (A4) states that past and future instances of $\{\boldsymbol{X}_t\}_{t\in\mathbb{Z}}$ are asymptotically independent. In terms of the functions $f_i$, the mixing condition roughly states that the noise variables play a non-negligible role. Assumption (A4) is thus similar to the structural minimality assumption from the causal inference literature [Bongers et al., 2021, Definition 2.10]. In comparison to other mixing conditions, Assumption (A4) is rather weak [Bradley, 2005]: Many other mixing conditions imply $\alpha$-strong mixing. Besides, as we will see in Section 4 and further discuss in Remark 6, $\alpha$-strong mixing is also implicitly assumed by Thams et al. [2022] if the noise variables have a density with respect to Lebesgue measure.

Establishing mixing conditions is rather difficult. However, there are several papers that prove mixing conditions for various processes [Ibragimov and Rozanov, 1978, Mokkadem, 1988, Chan and Tong, 1994, Xia and An, 1999, Cline and Pu, 1999, Bradley, 2005, among others].

**Assumption violations:** We now illustrate that one generally requires assumptions similar to (A3) and (A4) to ensure that a $d$-separation global Markov property holds: We present counterexamples for which just one assumption is violated while the other assumptions are satisfied. Doing so shows that none of Assumptions (A0)–(A4) alone suffices to establish a $d$-separation global Markov property.[8]

Violation of Assumption (A3):[9] Let $\phi \in \mathbb{R}$ with $|\phi| > 1$. Furthermore, let $\boldsymbol{\epsilon}_t := \{\epsilon_t^1, \epsilon_t^2\}_{t\in\mathbb{Z}}$ be a stochastic process satisfying (A0) and where each $\boldsymbol{\epsilon}_t$ has the same bivariate Gaussian distribution with mean zero and identity covariance matrix. Define the stochastic process $\{\boldsymbol{X}_t\}_{t\in\mathbb{Z}} := \{X_t^1, X_t^2\}_{t\in\mathbb{Z}}$ for each $t \in \mathbb{Z}$ by

$$X_t^1 = -\sum_{i=1}^{\infty} \phi^{-i}(\epsilon_{t+i}^1 + \epsilon_{t+i-1}^2) \quad \text{and}$$
$$X_t^2 = \epsilon_t^2. \tag{3}$$

Note that $\{\phi^{-i}\}_{i\in\mathbb{N}_{\geq 1}}$ is absolutely summable and hence $X_t^1$, interpreted as a limit in mean-square, exists and is uniquely defined up to $P$-nullsets [Lütkepohl, 2005, Section C.3]. Also note that Assumption (A3) is violated: For

---

[8]For (A2), we are not able to write down such a counterexample — for the examples we could think of at least one of the other assumptions was also always violated (see Example 2 in the SM for one such example).

[9]See Example 3 in the SM for a numerical illustration of this assumption violation.

example, just consider the covariance between $X_t^1$ and $\epsilon_{t+1}^2$ which equals $-\phi^{-2}$ to see that $X_t^1$ and $\epsilon_{t+1}^2$ are dependent.

Some calculations (see Lemma 4 in the SM) show that $\{\boldsymbol{X}_t\}_{t\in\mathbb{Z}}$ and $\{\boldsymbol{\epsilon}_t\}_{t\in\mathbb{Z}}$ solve the stochastic difference equations

$$X_t^1 = \phi \cdot X_{t-1}^1 + X_{t-1}^2 + \epsilon_t^1 \quad \text{and}$$
$$X_t^2 = \epsilon_t^2 \tag{4}$$

$P$-almost surely for all $t \in \mathbb{Z}$. There are clearly no contemporaneous edges in the induced full time graph, so Assumption (A1) holds. To verify the other assumptions, we define time reversed processes $\{\tilde{\boldsymbol{X}}_t\}_{t\in\mathbb{Z}}$ and $\{\tilde{\boldsymbol{\epsilon}}_t\}_{t\in\mathbb{Z}}$ by $(\tilde{X}_t^1, \tilde{X}_t^2) = (X_{-t}^1, -X_{-t-1}^2/\phi)$ and $(\tilde{\epsilon}_t^1, \tilde{\epsilon}_t^2) = (-\epsilon_{-t+1}^1/\phi, -\epsilon_{-t-1}^2/\phi)$. Again, some calculations (see Lemma 4 in the SM) show that $\{\tilde{\boldsymbol{X}}_t\}_{t\in\mathbb{Z}}$ and $\{\tilde{\boldsymbol{\epsilon}}_t\}_{t\in\mathbb{Z}}$ satisfy the VAR stochastic difference equations

$$\tilde{X}_t^1 = \frac{1}{\phi}\tilde{X}_{t-1}^1 + \tilde{X}_{t-1}^2 + \tilde{\epsilon}_t^1 \quad \text{and}$$
$$\tilde{X}_t^2 = \tilde{\epsilon}_t^2$$

$P$-almost surely for all $t \in \mathbb{Z}$. Clearly, $\{\tilde{\boldsymbol{\epsilon}}_t\}_{t\in\mathbb{Z}}$ is bivariate Gaussian with mean zero and diagonal covariance matrix. Moreover, $\{\tilde{\boldsymbol{X}}_t\}_{t\in\mathbb{Z}}$ is a stable VAR process with independent Gaussian noise which implies that $\{\tilde{\boldsymbol{X}}_t\}_{t\in\mathbb{Z}}$ is strictly stationary [Lütkepohl, 2005, p.16 and Proposition 2.1]. Therefore, $\{\boldsymbol{X}_t\}_{t\in\mathbb{Z}}$ is strictly stationary and so Assumption (A2) is satisfied.

Furthermore, Mokkadem [1988] shows that $\{\tilde{\boldsymbol{X}}_t\}_{t\in\mathbb{Z}}$ is $\alpha$-strongly mixing, which, together with Assumption (A2), implies that $\{\boldsymbol{X}_t\}_{t\in\mathbb{Z}}$ is $\alpha$-strongly mixing, so Assumption (A4) is satisfied (see Lemmas 6 and 7 in the SM for a formal proof of this implication).

In the corresponding full time graph, $X_t^1$ and $X_{t+1}^2$ are $d$-separated by the empty set, however, $X_t^1$ depends on $X_{t+1}^2 = \epsilon_{t+1}^2$. Thus, the $d$-separation global Markov property for this solution of (4) and the full time graph does not hold.

Violation of Assumption (A4):

Take any two random variables $Z_1$ and $Z_2$ that are dependent and take any $\{\boldsymbol{\epsilon}_t\}_{t\in\mathbb{Z}}$ that satisfies Assumption (A0) and $\{\boldsymbol{\epsilon}_t\}_{t\in\mathbb{Z}} \perp\!\!\!\perp (Z_1, Z_2)$. Define $\{\boldsymbol{X}_t\}_{t\in\mathbb{Z}}$ by

$$X_t^1 = Z_1 \quad \text{and}$$
$$X_t^2 = Z_2.$$

One can see that $\{\boldsymbol{X}_t\}_{t\in\mathbb{Z}}$ and $\{\boldsymbol{\epsilon}_t\}_{t\in\mathbb{Z}}$ satisfy the stochastic difference equations

$$X_t^1 = X_{t-1}^1 \quad \text{and}$$
$$X_t^2 = X_{t-1}^2. \tag{5}$$

One can also see that Assumption (A1) holds because the induced full time graph clearly contains no contemporaneous

edges. Also, by construction, Assumptions (A2) and (A3) hold. However, Assumption (A4) does not hold because every $\alpha(m)$ is lower bounded by $\sup_{A \in \Sigma_1^X, B \in \Sigma_2^X} |P(Z_1 \in A, Z_2 \in B) - P(Z_1 \in A)P(Z_2 \in B)|$ which is positive because $Z_1$ and $Z_2$ are dependent.

In the corresponding full time graph, $X_t^1$ and $X_t^2$ are $d$-separated by the empty set, however, by construction, they are dependent. Thus, the $d$-separation global Markov property for this solution of (5) and the full time graph does not hold.

# 3 PROVING THE GLOBAL MARKOV PROPERTY

In **Section 3.1**, we define finite graphs that represent the stochastic processes $\{\boldsymbol{X}_t\}_{t \in \mathbb{Z}}$ and $\{(\boldsymbol{X}_t, \boldsymbol{\epsilon}_t)\}_{t \in \mathbb{Z}}$ in a given finite-length time interval. In the same section, we then show that the distribution of $\{\boldsymbol{X}_t\}_{t \in \mathbb{Z}}$ respectively $\{(\boldsymbol{X}_t, \boldsymbol{\epsilon}_t)\}_{t \in \mathbb{Z}}$ in this finite-length time interval is globally Markov in an $m$-separation sense with respect to the corresponding finite graph. Finally, we show in **Section 3.2** that $d$-separation in the (augmented) full time graph implies $m$-separation in these finite graphs.

Our proof thus works as follows: A $d$-separation statement in the (augmented) full time graph implies an $m$-separation statement in a certain finite graph (**Lemma 3**). Due to the $m$-separation global Markov property (**Lemma 2**), the respective conditional independence holds.

## 3.1 FINITE CUTOFF GRAPHS

We now define cutoff graphs and augmented cutoff graphs.

**Definition 3** ((Augmented) $(t_0, n)$-cutoff graph)**.** Write $q := \max_{i \in [d]_1} q_i$. For given $t_0 \in \mathbb{Z}$ and $n \in \mathbb{N}_{\geq 0}$, the $(t_0, n)$-cutoff graph is a finite graph with vertex set $\{X_t^i : t \in [t_0 - n - q, t_0] \cap \mathbb{Z}, i \in [d]_1\}$. The $(t_0, n)$-cutoff graph has directed and bidirected edges: There is a directed edge from $X_s^j$ to $X_t^i$ if and only if $t \geq t_0 - n$ and $X_s^j$ is a causal parent of $X_t^i$. Moreover, there is a bidirected edge between $X_s^j$ and $X_t^i$ if and only if $s, t < t_0 - n$ and $X_s^j$ and $X_t^i$ have a common ancestor in the full time graph (which might equal $X_s^j$ or $X_t^i$).

The augmented $(t_0, n)$-cutoff graph consists of further vertices $\{\epsilon_t^i : t \in [t_0 - n, t_0] \cap \mathbb{Z}, i \in [d]_1\}$ and a directed edge from $\epsilon_t^i$ to $X_t^i$ if the former is a causal parent of the latter.

**Remark 4.** *These cutoff graphs very much look like a finite version of the (augmented) full time graph: The directed edges in the (augmented) $(t_0, n)$-cutoff graph are exactly as the directed edges in the (augmented) full time graph that point to a vertex in $[t_0 - n, t_0]$. Only the edges with both endpoints in the interval $[t_0 - n - q, t_0 - n - 1]$ differ.*

We now introduce some terminology that applies both to $\{\boldsymbol{X}_t\}_{t \in \mathbb{Z}}$ and $\{\boldsymbol{\epsilon}_t\}_{t \in \mathbb{Z}}$: For given $t_0 \in \mathbb{Z}$ and $n \in \mathbb{N}_{\geq 0}$, we call the variables in the interval $[t_0 - n - q, t_0 - n - 1]$ *auxiliary* variables. Also, we call the variables in the interval $[t_0 - n, t_0]$ *proper* variables. Furthermore, we call the variables in the interval $(-\infty, t_0 - n - q - 1]$ *ancient* variables and in the interval $[t_0 + 1, \infty)$ *future* variables.

**Example 1.** (continued) Let $t_0 \in \mathbb{Z}$ and $n = 2$. The auxiliary variables that are represented in the (augmented) $(t_0, n)$-cutoff graph are $X_{t_0-4}^1, X_{t_0-3}^1, X_{t_0-4}^2$ and $X_{t_0-3}^2$. The proper variables that are represented in the (augmented) $(t_0, n)$-cutoff graph are $X_{t_0-2}^1, X_{t_0-1}^1, X_{t_0}^1, X_{t_0-2}^2, X_{t_0-1}^1, X_{t_0}^1$ (and $\epsilon_{t_0-2}^1, \epsilon_{t_0-1}^1, \epsilon_{t_0}^1, \epsilon_{t_0-2}^2, \epsilon_{t_0-1}^1, \epsilon_{t_0}^1$). The (augmented) $(t_0, 2)$-cutoff graph is as in Figure 2.

**Districts and dependence:** One can write every proper variable as a function of the auxiliary variables and noise variables. Thus, to understand $m$-separation between proper variables in a cutoff graph, it is crucial to understand $m$-separation between the auxiliary variables. Similarly, to understand the dependence structure between the proper variables, it is crucial to understand the dependence structure between the auxiliary variables.

To understand $m$-separation between the auxiliary variables, the notion of *districts* is important [Richardson, 2003, Section 3]. Districts are the connected components into which one can partition the auxiliary variables of a cutoff graph: Two auxiliary variables are in the same connected component if and only if there is a path between them in the respective cutoff graph just consisting of bidirected edges.

As an important sub-result, we first show that each district is independent of the other districts. As we will see later, this sub-result then enables us to prove a global $m$-separation Markov property for the (augmented) $(t_0, n)$-cutoff graph and the corresponding distribution. The following lemma leads us to this sub-result.

**Lemma 1** (Extended Reichenbach's common cause principle for (augmented) full time graphs)**.** *Let Assumptions (A0)–(A4) hold. Let $J_1, J_2 \subseteq [d]_1 \times \mathbb{Z}$ be finite disjoint index sets and define $\boldsymbol{S}_1 := \{X_t^i : (i, t) \in J_1\}$ and $\boldsymbol{S}_2 := \{X_t^i : (i, t) \in J_2\}$. If $\boldsymbol{S}_1$ and $\boldsymbol{S}_2$ are dependent, then $\boldsymbol{S}_1$ and $\boldsymbol{S}_2$ have a common ancestor in the full time graph (which might be in either $\boldsymbol{S}_1$ or $\boldsymbol{S}_2$).*

*Proof.* We here just provide a proof idea, for the details see Section B.2 in the SM.

Notational remark: For $\Delta \in \mathbb{N}_{\geq 0}$, let $\boldsymbol{S}_{1,\Delta}$ and $\boldsymbol{S}_{2,\Delta}$ be the sets of variables that are constructed by time-shifting all elements of $\boldsymbol{S}_1$ respectively $\boldsymbol{S}_2$ by $\Delta$ time steps to the future.

Proof by contraposition: Suppose that $\boldsymbol{S}_1$ and $\boldsymbol{S}_2$ are dependent but do not have a common ancestor in the full time

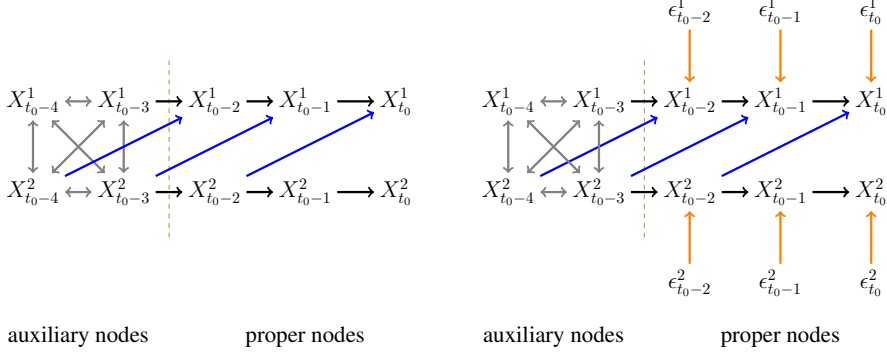

Figure 2: $(t_0, 2)$-cutoff graph (left) and augmented $(t_0, 2)$-cutoff graph (right) from Example 1.

graph. Now because of the acyclicity assumption (A1), for all $\Delta \in \mathbb{N}_{\geq 0}$, we can write $\boldsymbol{S}_{1,\Delta}$ and $\boldsymbol{S}_{2,\Delta}$ as functions of the auxiliary and noise variables (with respect to some $t_0 \in \mathbb{Z}$ and $n \in \mathbb{N}_{\geq 0}$ such that all elements of $\boldsymbol{S}_{1,\Delta}$ and $\boldsymbol{S}_{2,\Delta}$ are proper). Dependence between $\boldsymbol{S}_{1,\Delta}$ and $\boldsymbol{S}_{2,\Delta}$ can just be due to dependence between the auxiliary variables, because otherwise, either Assumption (A3) would be violated or $\boldsymbol{S}_{1,\Delta}$ and $\boldsymbol{S}_{2,\Delta}$, and hence, $\boldsymbol{S}_1$ and $\boldsymbol{S}_2$ would have a common ancestor in the full time graph, contradiction. Thus, schematically, the dependence between $\boldsymbol{S}_{1,\Delta}$ and $\boldsymbol{S}_{2,\Delta}$ is the dependence between $\boldsymbol{S}_{1,\Delta}$ and the auxiliary variables combined with the dependence between $\boldsymbol{S}_{2,\Delta}$ and the auxiliary variables. Now, $\alpha$-strong mixing implies that the dependence strength between $\boldsymbol{S}_{1,\Delta}$ respectively $\boldsymbol{S}_{2,\Delta}$ and the auxiliary variables, and thus, the dependence strength between $\boldsymbol{S}_{1,\Delta}$ and $\boldsymbol{S}_{2,\Delta}$ eventually decreases for large enough $\Delta$. That fact, however, is a contradiction, because by strict stationarity the dependence strength between $\boldsymbol{S}_{1,\Delta}$ and $\boldsymbol{S}_{2,\Delta}$ is the same as that between $\boldsymbol{S}_1$ and $\boldsymbol{S}_2$ for all $\Delta \in \mathbb{N}_{\geq 0}$. $\quad\square$

Lemma 1 immediately implies that each district is independent of the other districts: If this fact had not been true, then there would be a common ancestor between at least two districts in the (augmented) full time graph and hence, a bidirected edge between two districts in the (augmented) $(t_0, n)$-cutoff graph. This bidirected edge implies that these two separate districts are in the same district, contradiction.

Equipped with this fact about districts, we now prove an $m$-separation global Markov property for the (augmented) $(t_0, n)$-cutoff graph and the corresponding distribution.

**Lemma 2** (Global $m$-separation Markov property for the (augmented) $(t_0, n)$-cutoff graphs and corresponding distributions). *For all $t_0 \in \mathbb{Z}$ and for all $n \in \mathbb{N}_{\geq 0}$, a global $m$-separation Markov property holds for the $(t_0, n)$-cutoff graph and corresponding distribution of $\{\boldsymbol{X}_t\}_{t\in\mathbb{Z}}$. That is, for any finite disjoint index sets $J_1, J_2, J_3 \subseteq [d]_1 \times ([t_0 - n - q, t_0] \cap \mathbb{Z})$ such that $\boldsymbol{S}_3 := \{X_t^i : (i,t) \in J_3\}$ $m$-separates $\boldsymbol{S}_1 := \{X_t^i : (i,t) \in J_1\}$ and $\boldsymbol{S}_2 := \{X_t^i : (i,t) \in J_2\}$ in the $(t_0, n)$-cutoff graph, $\boldsymbol{S}_1 \perp\!\!\!\perp \boldsymbol{S}_2 \mid \boldsymbol{S}_3$.*

*An analogous result for the augmented $(t_0, n)$-cutoff graph and corresponding distribution of $\{(\boldsymbol{X}_t, \boldsymbol{\epsilon}_t)\}_{t\in\mathbb{Z}}$.*

*Proof.* We here again just provide a proof idea, for the details see Section B.3 in the SM.

The proof is similar to the proof for SCMs with mutually independent exogenous variables [Pearl, 2009, Janzing and Schölkopf, 2010]: First, we prove an $m$-separation local Markov property for the augmented $(t_0, n)$-cutoff graph [Richardson, 2003, Section 3]. This local Markov property requires us to show that a proper vertex is independent of its nondescendants without its parents given its parents. This case works as in the original SCM-proof and is rather immediate. Moreover, we need to show that an auxiliary vertex is independent of its nondescendants given the other auxiliary variables in its district (roughly speaking). As one can write the nondescendants of that auxiliary vertex as a function of the other auxiliary vertices, this independence question boils down to whether separate districts are independent of each other. Lemma 1 answers that question.

Now, this local Markov property implies an $m$-separation global Markov property for the augmented graph and corresponding distribution [Richardson, 2003, Theorem 2]. Finally, because each noise variable is only adjacent to at most one endogenous variable, we conclude that $m$-separation in the not-augmented graph implies $m$-separation in the augmented graph. From this fact, the $m$-separation global Markov property for the not-augmented graph follows. $\quad\square$

## 3.2 $d$-SEPARATION IN THE FULL TIME GRAPH IMPLIES $m$-SEPARATION IN THE FINITE CUTOFF-GRAPHS

In the following lemma, we show that $d$-separation in the (augmented) full time graph implies $m$-separation in all sufficiently large (augmented) cutoff graphs.

**Lemma 3.** *Let Assumptions (A0)–(A4) hold. For any finite disjoint index sets $J_1, J_2, J_3 \subseteq [d]_1 \times \mathbb{Z}$ such that $\boldsymbol{S}_3 :=$*

$\{X_t^i : (i,t) \in J_3\}$ *d-separates* $\boldsymbol{S}_1 := \{X_t^i : (i,t) \in J_1\}$ *and* $\boldsymbol{S}_2 := \{X_t^i : (i,t) \in J_2\}$ *in the full time graph, it holds that* $\boldsymbol{S}_3$ *m-separates* $\boldsymbol{S}_1$ *and* $\boldsymbol{S}_2$ *in all* $(t_0, n)$-*cutoff graphs for which all occuring vertices are proper.*

*An analogous result including noise variables* $\epsilon_t^i$ *holds true for the augmented full time graph and the augmented cutoff-graphs.*

*Proof.* The following proof both works for the non-augmented and augmented case: Let $t_0 \in \mathbb{Z}$ and $n \in \mathbb{N}_{\geq 0}$ be such that all elements of $\boldsymbol{S}_1$, $\boldsymbol{S}_2$ and $\boldsymbol{S}_3$ are proper.

Proof by contraposition: Assume that $\boldsymbol{S}_1$ and $\boldsymbol{S}_2$ are $m$-connected given $\boldsymbol{S}_3$ in the (augmented) $(t_0, n)$-cutoff graph. Then, there exists an $m$-connecting path $\pi^{t_0,n}$ between $\boldsymbol{S}_1$ and $\boldsymbol{S}_2$ given $\boldsymbol{S}_3$ in the (augmented) $(t_0, n)$-cutoff graph. We now construct a $d$-connecting path $\pi$ between $\boldsymbol{S}_1$ and $\boldsymbol{S}_2$ given $\boldsymbol{S}_3$ in the (augmented) full time graph: All directed edges and all vertices incident to these directed edges are copied from $\pi^{t_0,n}$ to $\pi$. All remaining auxiliary vertices in $\pi^{t_0,n}$ are also taken to $\pi$. The bidirected edges between two auxiliary vertices in $\pi^{t_0,n}$ are replaced by collider-free subpaths which only consist of auxiliary and ancient vertices (which is possible by construction of these bidirected edges); it does not matter which particular subpaths one takes.

The proper vertices in $\pi$ are distinct because the proper vertices in $\pi^{t_0,n}$ are distinct and because the collider-free subpaths that replace the bidirected edges do not contain proper vertices. However, the auxiliary and ancient vertices in $\pi$ are not necessarily distinct. To make $\pi$ a path, we repeatedly apply the following procedure until all vertices in $\pi$ are unique: Take some non-distinct auxiliary or ancient vertex $v$ in $\pi$ (the order of the non-distinct vertices does not matter). Delete all vertices and corresponding edges after the first occurence of $v$ in $\pi$ until the last occurence of $v$ in $\pi$. Then, stitch together the two remaining parts of $\pi$. That is, if $u{\ast}{-}{\ast}v$ is the edge corresponding to the first occurence of $v$ and $v{\ast}{-}{\ast}w$ is the edge corresponding to the last occurence of $v$, then $\pi$ will contain the triplet $u{\ast}{-}{\ast}v{\ast}{-}{\ast}w$ with all vertices and edges in between deleted.

If $v$ is a collider in $u{\ast}{-}{\ast}v{\ast}{-}{\ast}w$, so $u \to v \leftarrow w$, then $v$ is an ancestor of a vertex $\tilde{v}$ in the (augmented) full time graph such that $\tilde{v}$ is a collider in $\pi^{t_0,n}$: The vertex $v$ is only non-unique in the initial (so before deletion) $\pi$ if it occurs on at least two of the collider-free common ancestor subpaths. Consider the two collider-free common ancestor subpaths corresponding to the first and last occurence of $v$ in the initial $\pi$, and consider the two corresponding bidirected edges $\tilde{u}_1 \leftrightarrow \tilde{u}_2$ and $\tilde{w}_1 \leftrightarrow \tilde{w}_2$ (see Figure 3a for a schematic illustration). Without loss of generality, assume that $\pi^{t_0,n}$ first visits $\tilde{u}_2$, then $\tilde{u}_1$, then $\tilde{w}_1$ (which might equal $\tilde{u}_1$) and then $\tilde{w}_2$. Suppose now that $v$ is not an ancestor of $\tilde{u}_1$ in the (augmented) full time graph. Then, on the collider-free subpath from $\tilde{u}_2$ to $\tilde{u}_1$, the initial $\pi$ first comes to $v$ and then

to $u$ (Figure 3b), because otherwise, there would either not be the edge $u \to v$ or $v$ would be an ancestor of $\tilde{u}_1$ (note that both $v$ and $u$ occur *exactly* once on the collider-free subpath between $\tilde{u}_2$ and $\tilde{u}_1$ because that subpath is a *path*). Thus, the last edge before the first occurence of $v$ is not $u \to v$ (in Figure 3b it is $v \to r \neq u$) and hence, after deletion, the triplet $u{\ast}{-}{\ast}v{\ast}{-}{\ast}w$ cannot occur in $\pi$, contradiction. Thus, $v$ is an ancestor of $\tilde{u}_1$ (Figure 3c). In $\pi^{t_0,n}$, the vertex $\tilde{u}_1$ is either incident to another bidirected edge (Figure 3d), in which case it is a collider, or it is connected to $\tilde{w}_1$ via a path containing proper vertices and thus, at least one collider (Figure 3e). In the first case, let $\tilde{v} := \tilde{u}_1$, in the second case, let $\tilde{v}$ be the first collider on $\pi^{t_0,n}$ occuring after $\tilde{u}_1$.

We now show that no triplet $a{\ast}{-}{\ast}b{\ast}{-}{\ast}c$ in $\pi$ is $d$-blocked by $\boldsymbol{S}_3$.

**Case 1:** $b$ **is a proper vertex:** In this case, $a$ and $c$ are proper or auxiliary vertices as there are no edges from ancient to proper vertices. Thus, $a$, $b$ and $c$ exist in the (augmented) $(t_0, n)$-cutoff graph. The edges between $a$, $b$ and $c$ in the (augmented) full time graph are exactly as the edges between $a$, $b$ and $c$ in the (augmented) $(t_0, n)$-cutoff graph because edges incident to a proper vertex are always directed. Also, $a{\ast}{-}{\ast}b{\ast}{-}{\ast}c$ occurs in $\pi^{t_0,n}$ by construction of $\pi$. Hence, $b$ is a collider in this triplet in $\pi$ if and only if $b$ is a collider in this triplet in $\pi^{t_0,n}$.

If $b$ is a collider in this triplet in $\pi^{t_0,n}$, then a descendant of $b$ in the (augmented) $(t_0, n)$-cutoff graph is in $\boldsymbol{S}_3$. Hence, a descendant of $b$ in the (augmented) full time graph is in $\boldsymbol{S}_3$. If $b$ is a non-collider in this triplet in $\pi^{t_0,n}$, then $b \notin \boldsymbol{S}_3$.

**Case 2:** $b$ **is an auxiliary or ancient vertex:** Note that $\boldsymbol{S}_3$ only contains proper vertices, so $b \notin \boldsymbol{S}_3$. Thus, if $b$ is a non-collider in $a{\ast}{-}{\ast}b{\ast}{-}{\ast}c$ in $\pi$, then $a{\ast}{-}{\ast}b{\ast}{-}{\ast}c$ is not $d$-blocked by $\boldsymbol{S}_3$. If $b$ is a collider in $a{\ast}{-}{\ast}b{\ast}{-}{\ast}c$ in $\pi$, then $a$ and $c$ are also auxiliary or ancient. Moreover, $b$ is either at the end-respectively startpoint of two collider-free common ancestor subpaths, in which case $b$ is incident to two bidirected edges and hence a collider in $\pi^{t_0,n}$ (see Figure 4a), or $b$ has been introduced when applying the above-mentioned procedure for removing non-unique vertices (see Figure 4b). In the first case, $b$ has a descendant in $\boldsymbol{S}_3$ in the (augmented) $(t_0, n)$-cutoff graph and thus in the (augmented) full time graph. In the latter case, as already illustrated in Figure 3 and the corresponding paragraph, $b$ is an ancestor of a vertex $\tilde{b}$ in the (augmented) full time graph such that $\tilde{b}$ is a collider in $\pi^{t_0,n}$. Because $\tilde{b}$ has a descendant in $\boldsymbol{S}_3$ in the (augmented) $(t_0, n)$-cutoff graph, $\tilde{b}$ and hence $b$ has a descendant in $\boldsymbol{S}_3$ in the (augmented) full time graph. $\square$

## 4   APPLICATION TO VAR PROCESSES

In this section, we illustrate that (A0)–(A4) hold for stable finite-order VAR processes with independent finite-second-moment noise that has a density with respect to Lebesgue

measure (in Remark 6 we compare these assumptions to Thams et al. [2022]). For the basics on VAR processes, we refer to Chapter 2 and Section C.3 of Lütkepohl [2005].

Let $q \in \mathbb{N}_{\geq 0}$ be the order of the VAR process. Let $\{\boldsymbol{\epsilon}_t\}_{t \in \mathbb{Z}}$ be a stochastic process satisfying (A0) and for which $P_{\boldsymbol{\epsilon}_t}$ has mean zero, finite second moments, and is absolutely continuous with respect to Lebesgue measure. Moreover, consider $\mathbb{R}^{d \times d}$-matrices $\boldsymbol{A}^{(1)}, \ldots, \boldsymbol{A}^{(q)}$ which satisfy the *stability condition*, that is, for which $\det(\boldsymbol{I_d} - \boldsymbol{A}^{(1)}\lambda - \boldsymbol{A}^{(2)}\lambda^2 - \cdots - \boldsymbol{A}^{(q)}\lambda^p) = 0$ implies $|\lambda| > 1$. Now, let $\{\boldsymbol{X}_t\}_{t \in \mathbb{Z}}$ be a stochastic process such that

$$\boldsymbol{X}_t = \boldsymbol{A}^{(1)}\boldsymbol{X}_{t-1} + \ldots + \boldsymbol{A}^{(q)}\boldsymbol{X}_{t-q} + \boldsymbol{\epsilon}_t \qquad (6)$$

$P$-almost surely for all $t \in \mathbb{Z}$.

By definition, $\{\boldsymbol{X}_t\}_{t \in \mathbb{Z}}$ is a stable VAR process and thus has a moving average (MA) representation which just depends on past and present noise terms: That is, one can write $\boldsymbol{X}_t = \sum_{i=0}^{\infty} \boldsymbol{\Phi}_i \boldsymbol{\epsilon}_{t-i}$ where each $\boldsymbol{\Phi}_i \in \mathbb{R}^{d \times d}$ and such that $\{(\boldsymbol{\Phi}_i)_{k,l}\}_{i \in \mathbb{N}_{\geq 0}}$ is absolutely summable for all $k, l \in [d]_1$. Here, the infinite sum is meant as a limit in mean-square which is uniquely defined up to $P$-nullsets.

This MA representation allows us to verify Assumption (A2): By Assumption (A0), for all finite index sets $J \subseteq \mathbb{Z}$ and for all $N \in \mathbb{N}_{\geq 0}$, the joint distribution of $\{\boldsymbol{\epsilon}_{t-i} : t \in J, i \in [N]_0\}$ is time-shift invariant. Therefore, for all finite index sets $J \subseteq \mathbb{Z}$ and for all $N \in \mathbb{N}_{\geq 0}$, the distribution of $\{\sum_{i=0}^{N} \boldsymbol{\Phi}_i \boldsymbol{\epsilon}_{t-i} : t \in J\}$ is also time-shift invariant. Because these individual sums converge in mean-square, $\{\sum_{i=0}^{N} \boldsymbol{\Phi}_i \boldsymbol{\epsilon}_{t-i} : t \in J\}$ also converges in mean square and thus in distribution to $\{\boldsymbol{X}_t : t \in J\}$. Therefore, the distribution of $\{\boldsymbol{X}_t : t \in J\}$ is also time-shift invariant (see Lemma 9 in the SM for a formal proof of this implication).

The MA representation also allows us to verify Assumption (A3): By Assumption (A0), for all finite index sets $J \subseteq \mathbb{Z}$ and for all $N \in \mathbb{N}_{\geq 0}$, the set $\{\boldsymbol{\epsilon}_{t-i} : t \in J, i \in [N]_0\}$ is independent of any set of $\boldsymbol{\epsilon}_t$-variables that lie strictly in its future. Thus, also $\{\sum_{i=0}^{N} \boldsymbol{\Phi}_i \boldsymbol{\epsilon}_{t-i} : t \in J\}$ is independent of any set of $\boldsymbol{\epsilon}_t$-variables that all lie strictly in its future. By Lemma 8 in the SM, the same also holds for $\{\boldsymbol{X}_t : t \in J\}$.

Assumptions (A1) and (A4) are more immediate: (A1) follows because the full time graph does not have contemporaneous edges. (A4) follows from Mokkadem [1988].

**Remark 5.** *There are stationary nonstable VAR processes for which the global Markov property does not hold — just recall our example for a violation of Assumption (A3) from Section 2.3. To see that this 2-dimensional VAR(1) process is not stable, note that*

$$\boldsymbol{A}^{(1)} = \begin{pmatrix} \phi & 1 \\ 0 & 0 \end{pmatrix}$$

*and thus, $\det(\boldsymbol{I_d} - \boldsymbol{A}^{(1)}\lambda) = (1 - \lambda\phi)$. Because $(1 - \lambda\phi)$ has the root $\lambda = 1/\phi$ and because $|\phi| > 1$ by assumption, this VAR(1) process is not stable.*

**Remark 6.** *The global Markov property for VAR processes from Thams et al. [2022] requires stability and finite second moments and independence of the noise variables, as we do. As explained in Section 4 above, these assumptions together with equation* (6) *imply Assumptions (A0)–(A3).*

*However, Thams et al. [2022] do not require that $P_{\boldsymbol{\epsilon}_t}$ is absolutely continuous with respect to Lebesgue measure, which we do (but only for VAR processes and not for our general result!). We only need absolute-continuity to establish $\alpha$-strong mixing (note that there are stable VAR processes for which this absolute-continuity assumption does not hold and which are not strongly mixing [Doukhan, 1994, Section 2.3]). Thams et al. [2022] get around this absolute-continuity assumption because they directly employ the moving average representation of VAR processes in their proof. For nonlinear processes, such a moving average representation is typically not available, and a natural generalization are mixing assumptions [Fan and Yao, 2003, Section 2.6]. Our absolute-continuity assumption for VAR processes thus seems to be inherent to the more general nonlinear setting we consider.*

# 5 CONCLUSION AND OUTLOOK

In this paper, we proved a $d$-separation global Markov property for certain solutions of stochastic difference equations and the corresponding (augmented) full time graphs when $t \in \mathbb{Z}$. To the best of our knowledge, we presented the most general version of this result so far. The assumptions we made are natural and typical in the causal inference and time series literature. Furthermore, we illustrated why such assumptions are needed by investigating assumption violations. Our work is thus relevant whenever one causally models time series with $t \in \mathbb{Z}$ — a case that naturally arises when modelling equilibriums, for example.

In the future, loosening the assumptions, especially strict stationarity (A2), would be good. Here, we imagine that our result for the stationary case helps in establishing results for the nonstationary case as nonstationarity is often reduced to the stationary case (e.g., by assuming piecewise stationarity or by considering transformations of the nonstationary time series). One could also study the notion of faithfulness or even introduce the notion of time series SCMs and define what interventions and counterfactuals mean, similar in spirit to Bongers et al. [2021].

### Acknowledgements

We thank the anonymous reviewers for their helpful feedback and suggestions. J.R. received funding from the European Research Council (ERC) Starting Grant CausalEarth under the European Union's Horizon 2020 research and innovation program (Grant Agreement No. 948112) and J.R. also from No. 101003469 (XAIDA).

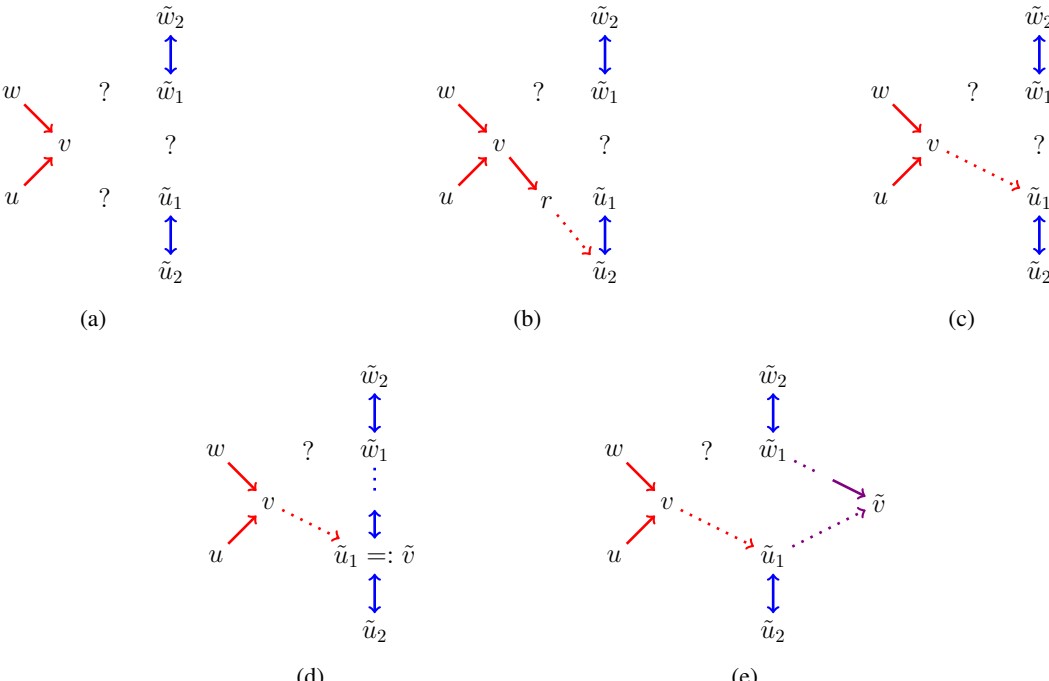

(a)  (b)  (c)

(d)  (e)

Figure 3: Schematic illustrations for the proof of Lemma 3: These schematic plots illustrate bidirected edges in the cutoff-graph and the snippets in the full time graph that replace them. Here, red edges/paths just occur in the (augmented) full time graph, blue edges/paths just in the (augmented) cutoff graph, and purple edges/paths appear in both. Also, ⋯⋯> stands for directed paths, ⋯ stands for paths, ⋯ → stands for a path where the last edge is → and ⋯ ↔ stands for a path where the last edge is ↔. A question mark indicates that the corresponding graphical structure around it is unknown. More detailed explanations are given in the proof.

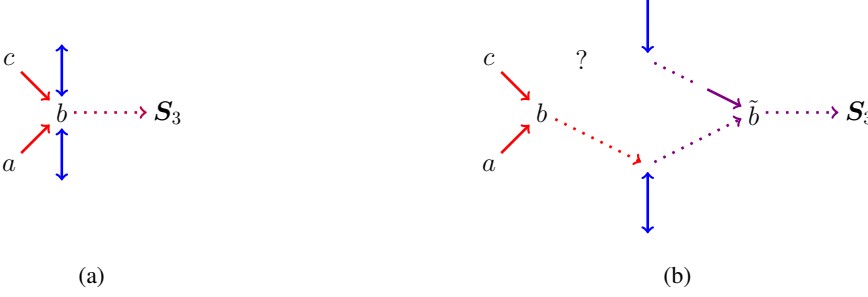

(a)  (b)

Figure 4: Schematic illustration for Case 2 in the proof of Lemma 3. The notation is as in Figure 3. Further explanations are again given in the proof.

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

# A Global Markov Property for Solutions of Stochastic Difference Equations and the corresponding Full Time Graphs
## (Supplementary Material)

**Tom Hochsprung**[1]  **Jakob Runge**[*1,2]  **Andreas Gerhardus**[*1]

[1]German Aerospace Center (DLR), Institute of Data Science, Jena, Germany
[2]Technische Universität Berlin, Berlin, Germany

## A   BASIC GRAPHICAL DEFINITIONS

In this section, we review some basic graphical notions that we use throughout paper. We mainly follow Richardson and Spirtes [2002], Richardson [2003] and Foygel et al. [2012].

**Mixed graphs:** A mixed graph $\mathcal{G}$ is a tuple $(V, D, B)$ where $V$ is a set of countably many vertices and $D, B \subseteq V \times V$ are the sets of directed and bidirected edges, respectively. For the set of directed edges $D$, we identify $(v, w) \in D$ with $v \to w$. Note that $(v, w) \in D$ does not necessarily imply $(w, v) \in D$. For the set of bidirected edges, $(v, w) \in B$ if and only if $(w, v) \in B$ and we identify these two tuples with the bidirected edge $v \leftrightarrow w$. There are no self-loops in either the directed or bidirected edge set, that is for all $v \in V$, it holds that $(v, v) \notin D \cup B$. For an edge $e = v \to w$ or $e = v \leftrightarrow w$, we say that $v$ and $w$ are incident to $e$ and that $e$ has endpoints $v$ and $w$. If two vertices $v$ and $w$ are connected by a directed or bidirected edge, then we say that $v$ and $w$ are adjacent. For a directed edge $v \to w$, we also say that $v$ points towards $w$. For a subset of vertices $A \subseteq V$ and a graph $G$, the induced subgraph $G_A$ has vertex set $A$ and all edges from $G$ with both endpoints in $A$. If there is the edge $v \to w$, then we call $v$ a parent of $w$ and $w$ a child of $v$. For a vertex $v$, we abbreviate the set of all its parents and all its childs by $\mathrm{pa}(v)$ and $\mathrm{ch}(v)$, respectively. For a set of vertices $A \subseteq V$, we define $\mathrm{pa}(A) := \bigcup_{v \in A} \mathrm{pa}(v)$ and $\mathrm{ch}(A) := \bigcup_{v \in A} \mathrm{ch}(v)$.

**Walks and paths:** A walk $\pi$ from a vertex $v$ to another vertex $w$ is a *finite* sequence of edges, that is, $\pi = (e_1, \ldots, e_n)$ for some $n \in \mathbb{N}_{\geq 0}$, such that there exists a sequence of (not necessarily distinct) vertices $(v = v_1, \ldots, v_{n+1} = w)$ such that the edge $e_i$ has endpoints $v_i$ and $v_{i+1}$. If all vertices on $\pi$ are distinct, then we call $\pi$ a path. If $n = 0$, then we say that $\pi$ is trivial. A path $\pi$ is called directed path if $e_1, \ldots, e_n$ are directed and point from $v_i$ to $v_{i+1}$. A path $\pi$ is called a directed cycle if $\pi$ is a non-trivial directed path with $v_1 = v_{n+1}$. For two sets of vertices $A, B \subseteq V$, we say that there is a path between them if there is a vertex $v \in A$ and a vertex $w \in B$ such that there is a path between $v$ and $w$.

**Ancestors and descendants:** A vertex $v$ is called an ancestor of another vertex $w$ and $w$ is called a descendant of $v$ if $v = w$ or if there is a directed path from $v$ to $w$. The set of all ancestors and descendants of a vertex $v$ is denoted by $\mathrm{an}(v)$ and $\mathrm{dec}(v)$, respectively. For a set of vertices $A \subseteq V$, we write $\mathrm{an}(A) := \bigcup_{v \in A} \mathrm{an}(v)$ and $\mathrm{dec}(A) := \bigcup_{v \in A} \mathrm{dec}(v)$. The set of nondescendants $\mathrm{nd}(A)$ equals $V \setminus \mathrm{dec}(A)$. We say that two sets of vertices $A$ and $B$ have a common ancestor if there is a vertex $v$ that both has a descendant in $A$ and a descendant in $B$.

**Types of graphs:** A mixed graph is called acyclic directed mixed graph (ADMG) if it does not contain directed cycles. If a mixed graph does not contain bidirected edges, that is $B = \emptyset$, then in slight abuse of notation, we identify it with the tuple $(V, D)$ instead of $(V, D, \emptyset)$ and call the graph a directed graph. If a directed graph does not contain directed cycles, then we call it a directed acyclic graph (DAG).

**Separation:** Consider a given ADMG. A vertex $v$ in a path $\pi$ is a collider (in $\pi$) if it is not an endpoint of $\pi$ and if the edges in $\pi$ to which $v$ is incident to are either $\to v \leftarrow$ or $\to v \leftrightarrow$ or $\leftrightarrow v \leftarrow$ or $\leftrightarrow v \leftrightarrow$. We call $v$ a non-collider in $\pi$ if it is not an endpoint of $\pi$ and not a collider in $\pi$. We say that a path $\pi$ between two vertices $v$ and $w$ is $m$-connecting given a third set of vertices $C$ for which $v, w \notin C$ if

- (Condition 1): every collider on $\pi$ has a descendant in $C$ and if

- (Condition 2): every non-collider on $\pi$ is not in $C$.

We say that a triplet $u\!*\!\!-\!\!*v\!*\!\!-\!\!*w$ in $\pi$ is $m$-connected given $C$ if both (Condition 1) and (Condition 2) hold, otherwise, we call that triplet $m$-blocked given $C$. We say that two sets of vertices $A$ and $B$ are $m$-connected given $C$ if there is an $m$-connecting path between them given $C$. If there is no $m$-connecting path $\pi$ between $A$ and $B$ given $C$, then we say that $A$ and $B$ are $m$-separated given $C$. If the ADMG is a DAG, then we call $m$-separation $d$-separation.

# B  PROOFS

## B.1  FURTHER NOTATION

For a system of sets $\mathcal{M}$, we write $\sigma(\mathcal{M})$ for the generated $\sigma$-algebra of $\mathcal{M}$. For random vectors $\boldsymbol{X}_1, \boldsymbol{X}_2, \ldots$, we write $\sigma(\boldsymbol{X}_1, \boldsymbol{X}_2, \ldots)$ for the generated $\sigma$-algebra thereof. For two $\sigma$-algebras $\Sigma_1$ and $\Sigma_2$, we write $\Sigma_1 \otimes \Sigma_2$ to denote the product $\sigma$-algebra of $\Sigma_1$ and $\Sigma_2$. For some fixed $n \in \mathbb{N}$, we write $\Sigma_1^{\otimes n}$ to denote the $n$-times product $\sigma$-algebra $\Sigma_1 \otimes \ldots \otimes \Sigma_1$. For two probability spaces $(\Omega_1, \mathcal{F}_1, P_1)$ and $(\Omega_2, \mathcal{F}_2, P_2)$, we write $P_1 \otimes P_2$ to denote the unique[1] product measure on $(\Omega_1 \times \Omega_2, \mathcal{F}_1 \otimes \mathcal{F}_2)$ that equals $P_1(E) \cdot P_2(F)$ for all rectangular sets $E \times F$ where $E \in \Omega_1$ and $F \in \Omega_2$.

## B.2  PROOF OF LEMMA 1

*Proof.* Notational remark: Let $t_{\min}(\boldsymbol{S}_1)$ and $t_{\max}(\boldsymbol{S}_1)$ be the minimal and maximal time indices of $\boldsymbol{S}_1$. Similarly, let $t_{\min}(\boldsymbol{S}_2)$ and $t_{\max}(\boldsymbol{S}_2)$ be the respective indices for $\boldsymbol{S}_2$. Moreover, write $(\mathcal{X}_{\boldsymbol{S}_1}, \Sigma_{\boldsymbol{S}_1})$ and $(\mathcal{X}_{\boldsymbol{S}_2}, \Sigma_{\boldsymbol{S}_2})$ for the measurable spaces into which $\boldsymbol{S}_1$ and $\boldsymbol{S}_2$ map, respectively. Also, write $\boldsymbol{S}_{1,\Delta}$ and $\boldsymbol{S}_{2,\Delta}$ for the time-shifted versions of $\boldsymbol{S}_1$ and $\boldsymbol{S}_2$, respectively: That is, the sets $\boldsymbol{S}_{1,\Delta}$ and $\boldsymbol{S}_{2,\Delta}$ contain all variables from the original sets $\boldsymbol{S}_1$ and $\boldsymbol{S}_2$ time-shifted to the future by $\Delta$ time steps. Note that by construction and the fact that all time instances of a component time series map into the same measurable space, $(\mathcal{X}_{\boldsymbol{S}_{1,\Delta}}, \Sigma_{\boldsymbol{S}_{1,\Delta}}) = (\mathcal{X}_{\boldsymbol{S}_1}, \Sigma_{\boldsymbol{S}_1})$ and $(\mathcal{X}_{\boldsymbol{S}_{2,\Delta}}, \Sigma_{\boldsymbol{S}_{2,\Delta}}) = (\mathcal{X}_{\boldsymbol{S}_2}, \Sigma_{\boldsymbol{S}_2})$, so we will always write $(\mathcal{X}_{\boldsymbol{S}_1}, \Sigma_{\boldsymbol{S}_1})$ and $(\mathcal{X}_{\boldsymbol{S}_2}, \Sigma_{\boldsymbol{S}_2})$.

Proof by contradiction: Assume that $\boldsymbol{S}_1$ and $\boldsymbol{S}_2$ are dependent but have no common ancestor in the full time graph. Let

$$\varepsilon := \sup_{A \in \Sigma_{\boldsymbol{S}_1},\, B \in \Sigma_{\boldsymbol{S}_2}} \left| P\Big(\boldsymbol{S}_1 \in A,\ \boldsymbol{S}_2 \in B\Big) - P\Big(\boldsymbol{S}_1 \in A\Big) P\Big(\boldsymbol{S}_2 \in B\Big) \right|.$$

By assumption, $\boldsymbol{S}_1$ and $\boldsymbol{S}_2$ are dependent, so $\varepsilon > 0$. Because $\{\boldsymbol{X}_t\}_{t \in \mathbb{Z}}$ is $\alpha$-strongly mixing, $\alpha(m) \to 0$ for $m \to \infty$. Thus, there is an $M \in \mathbb{N}_{\geq 0}$ such that for all $m \in \mathbb{N}_{\geq M}$ it holds that $\alpha(m) < \varepsilon$. Now, choose some $\Delta \in \mathbb{N}_{\geq M}$. Furthermore, choose $n \in \mathbb{N}_{\geq 0}$ and $t_0 \in \mathbb{Z}$ such that $t_{\min}(\boldsymbol{S}_1) + \Delta = t_0$, such that $\min\{t_{\min}(\boldsymbol{S}_1), t_{\min}(\boldsymbol{S}_2)\} \geq t_0 - n$, and such that $n \geq \Delta$.

Because of the acyclicity assumption (A1), we can write $\boldsymbol{S}_{1,\Delta}$ and $\boldsymbol{S}_{2,\Delta}$ as functions of the auxiliary variables and noise variables, that is

$$\boldsymbol{S}_{1,\Delta} = g_1(\boldsymbol{X}_{\mathrm{aux}}, \boldsymbol{\epsilon}_{1,\to}), \quad \text{and}$$
$$\boldsymbol{S}_{2,\Delta} = g_2(\boldsymbol{X}_{\mathrm{aux}}, \boldsymbol{\epsilon}_{2,\to}).$$

Here, $\boldsymbol{X}_{\mathrm{aux}}$ stands for the auxiliary variables (with respect to $t_0$ and $n$) and $\boldsymbol{\epsilon}_{1,\to}$ and $\boldsymbol{\epsilon}_{2,\to}$ stand for all proper and future noise variables that are ancestors of $\boldsymbol{S}_{1,\Delta}$ respectively $\boldsymbol{S}_{2,\Delta}$ in the full time graph. Moreover, let $(\mathcal{X}_{\mathrm{aux}}, \Sigma_{\mathrm{aux}})$ and $(\mathcal{E}_{1,\to}, \Sigma_{1,\to})$ and $(\mathcal{E}_{2,\to}, \Sigma_{2,\to})$ denote the corresponding measurable spaces. The functions $g_1$ and $g_2$ are measurable with respect to $(\mathcal{X}_{\mathrm{aux}} \times \mathcal{E}_{1,\to}, \Sigma_{\mathrm{aux}} \otimes \Sigma_{1,\to})$ and $(\mathcal{X}_{\mathrm{aux}} \times \mathcal{E}_{2,\to}, \Sigma_{\mathrm{aux}} \otimes \Sigma_{2,\to})$, respectively, because $g_1$ and $g_2$ are each compositions of the measurable functions $f_i$.

Note that $\boldsymbol{\epsilon}_{1,\to} \perp\!\!\!\perp \boldsymbol{\epsilon}_{2,\to}$ because otherwise by Assumption (A0), there needs to be a noise variable that is both in $\boldsymbol{\epsilon}_{1,\to}$ and $\boldsymbol{\epsilon}_{2,\to}$, which would in turn imply that $\boldsymbol{S}_{1,\Delta}$ and $\boldsymbol{S}_{2,\Delta}$ have a common ancestor in the full time graph, which would imply that $\boldsymbol{S}_1$ and $\boldsymbol{S}_2$ have a common ancestor in the full time graph, contradiction.

Moreover, by Assumption (A3) and because $\boldsymbol{\epsilon}_{1,\to}$ and $\boldsymbol{\epsilon}_{2,\to}$ only contain proper and future variables and hence occur strictly after $\boldsymbol{X}_{\mathrm{aux}}$, it holds that $(\boldsymbol{\epsilon}_{1,\to}, \boldsymbol{\epsilon}_{2,\to}) \perp\!\!\!\perp \boldsymbol{X}_{\mathrm{aux}}$. By the decomposition property of conditional independence[2], it particularly also holds that $\boldsymbol{\epsilon}_{1,\to} \perp\!\!\!\perp \boldsymbol{X}_{\mathrm{aux}}$.

---

[1]For uniqueness see Theorem B in §35 in Halmos [1974].

[2]See, e.g., Section 1.5 in Studený [2018] for this and other properties of conditional independence.

Therefore,

$$\begin{aligned}
P_{\boldsymbol{\epsilon}_{2,\to},\boldsymbol{\epsilon}_{1,\to},\boldsymbol{X}_{\text{aux}}} &= P_{\boldsymbol{\epsilon}_{2,\to},\boldsymbol{\epsilon}_{1,\to}} \otimes P_{\boldsymbol{X}_{\text{aux}}} \\
&= P_{\boldsymbol{\epsilon}_{2,\to}} \otimes P_{\boldsymbol{\epsilon}_{1,\to}} \otimes P_{\boldsymbol{X}_{\text{aux}}} \\
&= P_{\boldsymbol{\epsilon}_{2,\to}} \otimes P_{\boldsymbol{\epsilon}_{1,\to},\boldsymbol{X}_{\text{aux}}},
\end{aligned}$$

and thus, $\boldsymbol{\epsilon}_{2,\to} \perp\!\!\!\perp (\boldsymbol{\epsilon}_{1,\to}, \boldsymbol{X}_{\text{aux}})$ and hence, $\boldsymbol{\epsilon}_{2,\to} \perp\!\!\!\perp (\boldsymbol{S}_{1,\Delta}, \boldsymbol{X}_{\text{aux}})$.

Now, note that

$$\begin{aligned}
&\sup_{A\in\Sigma_{\boldsymbol{S}_1},\, B\in\Sigma_{\boldsymbol{S}_2}} \left| P\Big(\boldsymbol{S}_{1,\Delta} \in A, \boldsymbol{S}_{2,\Delta} \in B\Big) - P\Big(\boldsymbol{S}_{1,\Delta} \in A\Big) P\Big(\boldsymbol{S}_{2,\Delta} \in B\Big) \right| \\
&= \sup_{A\in\Sigma_{\boldsymbol{S}_1},\, B\in\Sigma_{\boldsymbol{S}_2}} \left| P\Big(\boldsymbol{S}_{1,\Delta} \in A,\ g_2(\boldsymbol{X}_{\text{aux}}, \boldsymbol{\epsilon}_{2,\to}) \in B\Big) \right. \\
&\qquad\qquad\qquad \left. - P\Big(\boldsymbol{S}_{1,\Delta} \in A\Big) P\Big(g_2(\boldsymbol{X}_{\text{aux}}, \boldsymbol{\epsilon}_{2,\to}) \in B\Big) \right| \\
&= \sup_{A\in\Sigma_{\boldsymbol{S}_1},\, B\in\Sigma_{\boldsymbol{S}_2}} \left| P\Big(\boldsymbol{S}_{1,\Delta} \in A, (\boldsymbol{X}_{\text{aux}}, \boldsymbol{\epsilon}_{2,\to}) \in g_2^{-1}(B)\Big) \right. \\
&\qquad\qquad\qquad \left. - P\Big(\boldsymbol{S}_{1,\Delta} \in A\Big) P\Big((\boldsymbol{X}_{\text{aux}}, \boldsymbol{\epsilon}_{2,\to}) \in g_2^{-1}(B)\Big) \right| \\
&= \sup_{\substack{A\in\Sigma_{\boldsymbol{S}_1},\, B'\in\Sigma_{\text{aux}}\otimes\Sigma_{2,\to}: \\ \exists B\in\Sigma_{\boldsymbol{S}_2}:\, g_2^{-1}(B)=B'}} \left| P\Big(\boldsymbol{S}_{1,\Delta} \in A, (\boldsymbol{X}_{\text{aux}}, \boldsymbol{\epsilon}_{2,\to}) \in B'\Big) \right. \\
&\qquad\qquad\qquad\qquad \left. - P\Big(\boldsymbol{S}_{1,\Delta} \in A\Big) P\Big((\boldsymbol{X}_{\text{aux}}, \boldsymbol{\epsilon}_{2,\to}) \in B'\Big) \right| \\
&\leq \sup_{A\in\Sigma_{\boldsymbol{S}_1},\, B'\in\Sigma_{\text{aux}}\otimes\Sigma_{2,\to}} \left| P\Big(\boldsymbol{S}_{1,\Delta} \in A, (\boldsymbol{X}_{\text{aux}}, \boldsymbol{\epsilon}_{2,\to}) \in B'\Big) \right. \\
&\qquad\qquad\qquad\qquad \left. - P\Big(\boldsymbol{S}_{1,\Delta} \in A\Big) P\Big((\boldsymbol{X}_{\text{aux}}, \boldsymbol{\epsilon}_{2,\to}) \in B'\Big) \right|.
\end{aligned}$$

Here, the second last equality follows because $g_2$ is measurable and thus $\{g_2^{-1}(B) : B \in \Sigma_{\boldsymbol{S}_2}\} \subseteq \Sigma_{\text{aux}} \otimes \Sigma_{2,\to}$.

For an arbitrary $y \in \mathcal{E}_{2,\to}$, define the $y$-section of $B'$ by $B'_y := \{x \in \mathcal{X}_{\text{aux}} :\ (x,y) \in B'\}$. A standard result in measure theory (see, e.g., Theorem A in §34 in Halmos [1974]) states that if $B' \in \Sigma_{\text{aux}} \otimes \Sigma_{2,\to}$, then $B'_y \in \Sigma_{\text{aux}}$ for all $y \in \mathcal{E}_{2,\to}$. Now, because $\boldsymbol{\epsilon}_{2,\to} \perp\!\!\!\perp (\boldsymbol{S}_{1,\Delta}, \boldsymbol{X}_{\text{aux}})$, we have $P_{\boldsymbol{S}_{1,\Delta},\boldsymbol{X}_{\text{aux}},\boldsymbol{\epsilon}_{2,\to}} = P_{\boldsymbol{S}_{1,\Delta},\boldsymbol{X}_{\text{aux}}} \otimes P_{\boldsymbol{\epsilon}_{2,\to}}$. Moreover, recall that probability measures are $\sigma$-finite measures. Therefore, we can apply Theorem B in §35 in Halmos [1974] which yields

$$\begin{aligned}
&P\Big(\boldsymbol{S}_{1,\Delta} \in A, (\boldsymbol{X}_{\text{aux}}, \boldsymbol{\epsilon}_{2,\to}) \in B'\Big) - P\Big(\boldsymbol{S}_{1,\Delta} \in A\Big) P\Big((\boldsymbol{X}_{\text{aux}}, \boldsymbol{\epsilon}_{2,\to}) \in B'\Big) \\
&= \int P_{\boldsymbol{S}_{1,\Delta},\boldsymbol{X}_{\text{aux}}}(A \times B'_y) dP_{\boldsymbol{\epsilon}_{2,\to}}(y) - \int P_{\boldsymbol{S}_{1,\Delta}}(A) P_{\boldsymbol{X}_{\text{aux}}}(B'_y) dP_{\boldsymbol{\epsilon}_{2,\to}}(y) \\
&= \int \Big(P_{\boldsymbol{S}_{1,\Delta},\boldsymbol{X}_{\text{aux}}}(A \times B'_y) - P_{\boldsymbol{S}_{1,\Delta}}(A) P_{\boldsymbol{X}_{\text{aux}}}(B'_y)\Big) dP_{\boldsymbol{\epsilon}_{2,\to}}(y).
\end{aligned}$$

Therefore,

$$
\sup_{A \in \Sigma_{\boldsymbol{S}_1}, \, B' \in \Sigma_{\mathrm{aux}} \otimes \Sigma_{2,\to}} \left| P\Big( \boldsymbol{S}_{1,\Delta} \in A, (\boldsymbol{X}_{\mathrm{aux}}, \boldsymbol{\epsilon}_{2,\to}) \in B' \Big) \right.
$$

$$
\left. - P\Big( \boldsymbol{S}_{1,\Delta} \in A \Big) P\Big( (\boldsymbol{X}_{\mathrm{aux}}, \boldsymbol{\epsilon}_{2,\to}) \in B' \Big) \right|
$$

$$
= \sup_{A \in \Sigma_{\boldsymbol{S}_1}, \, B' \in \Sigma_{\mathrm{aux}} \otimes \Sigma_{2,\to}} \left| \int \Big( P_{\boldsymbol{S}_{1,\Delta}, \boldsymbol{X}_{\mathrm{aux}}}(A \times B'_y) \right.
$$

$$
\left. - P_{\boldsymbol{S}_{1,\Delta}}(A) P_{\boldsymbol{X}_{\mathrm{aux}}}(B'_y) \Big) dP_{\boldsymbol{\epsilon}_{2,\to}}(y) \right|
$$

$$
\leq \sup_{A \in \Sigma_{\boldsymbol{S}_1}, \, B' \in \Sigma_{\mathrm{aux}} \otimes \Sigma_{2,\to}} \int \left| P_{\boldsymbol{S}_{1,\Delta}, \boldsymbol{X}_{\mathrm{aux}}}(A \times B'_y) \right.
$$

$$
\left. - P_{\boldsymbol{S}_{1,\Delta}}(A) P_{\boldsymbol{X}_{\mathrm{aux}}}(B'_y) \right| dP_{\boldsymbol{\epsilon}_{2,\to}}(y)
$$

$$
\leq \sup_{A \in \Sigma_{\boldsymbol{S}_1}, \, B' \in \Sigma_{\mathrm{aux}} \otimes \Sigma_{2,\to}} \int \sup_{A' \in \Sigma_{\boldsymbol{S}_1}, \, B'' \in \Sigma_{\mathrm{aux}}} \left| P_{\boldsymbol{S}_{1,\Delta}, \boldsymbol{X}_{\mathrm{aux}}}(A' \times B'') \right.
$$

$$
\left. - P_{\boldsymbol{S}_{1,\Delta}}(A') P_{\boldsymbol{X}_{\mathrm{aux}}}(B'') \right| dP_{\boldsymbol{\epsilon}_{2,\to}}(y)
$$

$$
\leq \sup_{A \in \Sigma_{\boldsymbol{S}_1}, \, B' \in \Sigma_{\mathrm{aux}} \otimes \Sigma_{2,\to}} \quad \sup_{A' \in \Sigma_{\boldsymbol{S}_1}, \, B'' \in \Sigma_{\mathrm{aux}}} \left| P_{\boldsymbol{S}_{1,\Delta}, \boldsymbol{X}_{\mathrm{aux}}}(A' \times B'') \right.
$$

$$
\left. - P_{\boldsymbol{S}_{1,\Delta}}(A') P_{\boldsymbol{X}_{\mathrm{aux}}}(B'') \right| \cdot \int dP_{\boldsymbol{\epsilon}_{2,\to}}(y)
$$

$$
= \sup_{A' \in \Sigma_{\boldsymbol{S}_1}, \, B'' \in \Sigma_{\mathrm{aux}}} \left| P_{\boldsymbol{S}_{1,\Delta}, \boldsymbol{X}_{\mathrm{aux}}}(A' \times B'') - P_{\boldsymbol{S}_{1,\Delta}}(A') P_{\boldsymbol{X}_{\mathrm{aux}}}(B'') \right|.
$$

By construction, every variable in $\boldsymbol{X}_{\mathrm{aux}}$ occurs before $t_0 - n$ and hence before $t_0 - \Delta$. Also by construction, every variable in $\boldsymbol{S}_{1,\Delta}$ occurs after $t_{\min}(\boldsymbol{S}_1) + \Delta$. Thus,

$$
\sup_{A' \in \Sigma_{\boldsymbol{S}_1}, \, B'' \in \Sigma_{\mathrm{aux}}} \left| P_{\boldsymbol{S}_{1,\Delta}, \boldsymbol{X}_{\mathrm{aux}}}(A' \times B'') - P_{\boldsymbol{S}_{1,\Delta}}(A') P_{\boldsymbol{X}_{\mathrm{aux}}}(B'') \right|.
$$

$$
\leq \alpha(t_{\min}(\boldsymbol{S}_1) + \Delta - (t_0 - \Delta))
$$

$$
= \alpha(t_0 - (t_0 - \Delta))
$$

$$
= \alpha(\Delta)
$$

$$
< \varepsilon.
$$

This inequality is a contradiction, because by strict stationarity (Assumption (A2)), the distribution of $(\boldsymbol{S}_1, \boldsymbol{S}_2)$ equals the distribution of $(\boldsymbol{S}_{1,\Delta}, \boldsymbol{S}_{2,\Delta})$ and thus,

$$
\sup_{A \in \Sigma_{\boldsymbol{S}_1}, \, B \in \Sigma_{\boldsymbol{S}_2}} \left| P\Big( \boldsymbol{S}_{1,\Delta} \in A, \boldsymbol{S}_{2,\Delta} \in B \Big) - P\Big( \boldsymbol{S}_{1,\Delta} \in A \Big) P\Big( \boldsymbol{S}_{2,\Delta} \in B \Big) \right|
$$

$$
= \varepsilon.
$$

$\square$

## B.3 PROOF OF LEMMA 2

*Proof.* Let $t_0 \in \mathbb{Z}$ and $n \in \mathbb{N}_{\geq 0}$ be arbitrary but fixed.

**Outline:**

To prove Lemma 2, we follow a similar outline as the proof for SCMs with mutually independent exogenous variables (see, for example, Lemma 2 in Janzing and Schölkopf [2010] and Theorem 1.4.1 in Pearl [2009]):

- First, we prove the ordered local $m$-separation Markov property [Richardson, 2003, Section 3] for the augmented $(t_0, n)$-cutoff graph and the corresponding distribution of $\{(\boldsymbol{X}_t, \boldsymbol{\epsilon}_t)\}_{t \in \mathbb{Z}}$ .
- Using Theorem 2 in Richardson [2003], we then obtain the global $m$-separation Markov property for the augmented $(t_0, n)$-cutoff graph and the corresponding distribution of $\{(\boldsymbol{X}_t, \boldsymbol{\epsilon}_t)\}_{t \in \mathbb{Z}}$.
- From the global $m$-separation Markov property for the augmented case, we then deduct the global $m$-separation Markov property for the not-augmented $(t_0, n)$-cutoff graph and the corresponding distribution of $\{\boldsymbol{X}_t\}_{t \in \mathbb{Z}}$.

**Notation and terminology:**

The notation and terminology in this proof largely follows Section 3 Richardson [2003]: Abbreviate the augmented $(t_0, n)$-cutoff graph by $\mathcal{M}^a$ and the not-augmented $(t_0, n)$-cutoff graph by $\mathcal{M}$. Write $\prec$ to denote a total ordering of the variables in $\mathcal{M}^a$ such that $x \prec y$ implies that $y \notin \mathbf{an}(x)$ (we call such an ordering consistent with respect to $\mathcal{M}^a$). Let

$$\mathbf{pre}_{\mathcal{M}^a, \prec}(x) := \{v \mid v \prec x \text{ or } v = x\}$$

and let the district of $x$ be

$$\mathbf{dis}_{\mathcal{M}^a}(x) := \{v \mid v \leftrightarrow \cdots \leftrightarrow x \text{ in } \mathcal{M}^a \text{ or } v = x\}.$$

Note that $\mathbf{dec}_{\mathcal{M}^a}(x) \cap \mathbf{pre}_{\mathcal{M}^a, \prec}(x) = \{x\}$.

Let $\boldsymbol{A}$ be an ancestral set of vertices, that is, $\mathbf{an}_{\mathcal{M}^a}(\boldsymbol{A}) = \boldsymbol{A}$. Let $x$ be a vertex in $\boldsymbol{A}$ such that no children of $x$ in $\mathcal{M}^a$ are in $\boldsymbol{A}$. Now, let $\mathcal{M}^a_{\boldsymbol{A}}$ denote the induced subgraph on $\boldsymbol{A}$. The Markov blanket of a vertex $x$ with respect to $\boldsymbol{A}$, is defined as

$$\mathbf{mb}_{\mathcal{M}^a}(x, \boldsymbol{A}) := \mathbf{pa}_{\mathcal{M}^a_{\boldsymbol{A}}}(\mathbf{dis}_{\mathcal{M}^a_{\boldsymbol{A}}}(x)) \cup (\mathbf{dis}_{\mathcal{M}^a_{\boldsymbol{A}}}(x) \setminus \{x\}).$$

The notation for the not-augmented graph $\mathcal{M}$ is analogous.

**Goal:**

We need to show that for all vertices $x$ in $\mathcal{M}^a$ and for any ancestral set $\boldsymbol{A} \subseteq \mathbf{pre}_{\mathcal{M}^a, \prec}(x)$ (which implies $\mathbf{ch}(A) \cap \boldsymbol{A} = \emptyset$) with $x \in \boldsymbol{A}$,

$$x \perp\!\!\!\perp \boldsymbol{A} \setminus (\mathbf{mb}_{\mathcal{M}^a}(x, \boldsymbol{A}) \cup \{x\}) \mid \mathbf{mb}_{\mathcal{M}^a}(x, \boldsymbol{A}).$$

**Main proof:**

Take any consistent ordering $\prec$ with respect to $\mathcal{M}^a$. Furthermore, take any ancestral set $\boldsymbol{A}$ such that $\boldsymbol{A} \subseteq \mathbf{pre}_{\mathcal{M}^a, \prec}(x)$ with $x \in \boldsymbol{A}$.

We now make the following case distinction.

**Case 1: $x$ is a proper non-noise vertex**

Because $x$ is a non-auxiliary vertex, we have $\mathbf{dis}_{\mathcal{M}^a}(x) = \{x\}$. Thus, $\mathbf{dis}_{\mathcal{M}^a_{\boldsymbol{A}}}(x) \subseteq \{x\}$. Because $x \in \boldsymbol{A}$, the vertex $x$ occurs in the subgraph $\mathcal{M}^a_{\boldsymbol{A}}$ and thus, $\mathbf{dis}_{\mathcal{M}^a_{\boldsymbol{A}}}(x) = \{x\}$.

Moreover, $\mathbf{pa}_{\mathcal{M}^a_{\boldsymbol{A}}}(x) \subseteq \mathbf{pa}_{\mathcal{M}^a}(x)$. Because $\boldsymbol{A}$ is ancestral and $x \in \boldsymbol{A}$, we have that $\mathbf{pa}_{\mathcal{M}^a}(x) \subseteq \boldsymbol{A}$, and hence all parents of $x$ occur as vertices in $\mathcal{M}^a_{\boldsymbol{A}}$. Therefore, $\mathbf{pa}_{\mathcal{M}^a_{\boldsymbol{A}}}(x) = \mathbf{pa}_{\mathcal{M}^a}(x)$. Hence, $\mathbf{mb}_{\mathcal{M}^a}(x, \boldsymbol{A}) = \mathbf{pa}_{\mathcal{M}^a}(x)$.

Furthermore,

$$\boldsymbol{A} \setminus \mathbf{pa}_{\mathcal{M}^a}(x) \subseteq \mathbf{pre}_{\mathcal{M}^a, \prec}(x) \setminus \mathbf{pa}_{\mathcal{M}^a}(x) \subseteq (\mathbf{nd}_{\mathcal{M}^a}(x) \cup \{x\}) \setminus \mathbf{pa}_{\mathcal{M}^a}(x). \tag{7}$$

We can write $x$ as a function of its parents (including the respective noise term). That is, for some measurable deterministic function $g$, we can write

$$x = g\big(\mathbf{pa}_{\mathcal{M}^a}(x)\big).$$

Here, measurability follows because $g$ equals one of the measurable $f_i$'s. Now, using a standard fact about conditional independence[3], we get

$$\mathbf{pa}_{\mathcal{M}^a}(x) \perp\!\!\!\perp \mathbf{nd}_{\mathcal{M}^a}(x) \setminus \mathbf{pa}_{\mathcal{M}^a}(x) \mid \mathbf{pa}_{\mathcal{M}^a}(x).$$

---

[3]This fact states that for random vectors $\boldsymbol{X}$ and $\boldsymbol{Y}$, it holds that $\boldsymbol{X} \perp\!\!\!\perp \boldsymbol{Y} \mid \boldsymbol{X}$, see, e.g., Definition 1.5.2 in Studený [2018] and the comment below this definition.

Therefore,

$$g\big(\mathbf{pa}_{\mathcal{M}^a}(x)\big) \perp\!\!\!\perp \mathbf{nd}_{\mathcal{M}^a}(x) \setminus \mathbf{pa}_{\mathcal{M}^a}(x) \mid \mathbf{pa}_{\mathcal{M}^a}(x),$$

and hence,

$$x \perp\!\!\!\perp \mathbf{nd}_{\mathcal{M}^a}(x) \setminus \mathbf{pa}_{\mathcal{M}^a}(x) \mid \mathbf{pa}_{\mathcal{M}^a}(x).$$

Thus, by (7), the decomposition property of conditional independence[4] and the fact that $\mathbf{mb}_{\mathcal{M}^a}(x, \mathbf{A}) = \mathbf{pa}_{\mathcal{M}^a}(x)$,

$$x \perp\!\!\!\perp \mathbf{A} \setminus (\mathbf{mb}_{\mathcal{M}^a}(x, \mathbf{A}) \cup \{x\}) \mid \mathbf{mb}_{\mathcal{M}^a}(x, \mathbf{A}).$$

### Case 2: $x$ is a proper noise vertex

If $x$ is a noise vertex, then $x$ does not have a parent in $\mathcal{M}^a$, so $\mathbf{pa}_{\mathcal{M}^a}(x) = \emptyset$. Moreover, $\mathbf{dis}_{\mathcal{M}^a}(x) = \{x\}$. Thus, $\mathbf{mb}_{\mathcal{M}^a}(x, \mathbf{A}) = \emptyset$.

One can write the nondescendants of $x$ in $\mathcal{M}^a$ as a deterministic function $g$ of the auxiliary variables $s_{\mathrm{aux}}$ and the remaining noise variables $s_\epsilon$ excluding $x$, that is $\mathbf{nd}_{\mathcal{M}^a}(x) = g(s_{\mathrm{aux}}, s_\epsilon)$. Note that $g$ is measurable because it is a composition of the measurable $f_i$'s. By Assumptions (A0) and (A3),

$$\begin{aligned}
P_{x, s_\epsilon, s_{\mathrm{aux}}} &= P_{x, s_\epsilon} \otimes P_{s_{\mathrm{aux}}} \\
&= P_x \otimes P_{s_\epsilon} \otimes P_{s_{\mathrm{aux}}} \\
&= P_x \otimes P_{s_\epsilon, s_{\mathrm{aux}}}
\end{aligned}$$

and thus,

$$x \perp\!\!\!\perp (s_{\mathrm{aux}}, s_\epsilon).$$

Therefore,

$$x \perp\!\!\!\perp g(s_{\mathrm{aux}}, s_\epsilon),$$

and hence,

$$x \perp\!\!\!\perp \mathbf{nd}_{\mathcal{M}^a}(x).$$

Because $\mathbf{A} \subseteq \mathbf{pre}_{\mathcal{M}^a, \prec}(x) \subseteq \mathbf{nd}_{\mathcal{M}^a}(x) \cup \{x\}$, the decomposition property of conditional independence together with $\mathbf{mb}_{\mathcal{M}^a}(x, \mathbf{A}) = \emptyset$ implies

$$x \perp\!\!\!\perp \mathbf{A} \setminus (\mathbf{mb}_{\mathcal{M}^a}(x, \mathbf{A}) \cup \{x\}) \mid \mathbf{mb}_{\mathcal{M}^a}(x, \mathbf{A}).$$

### Case 3: $x$ is an auxiliary vertex

Every vertex in the district of $x$ has no parents; so $\mathbf{mb}_{\mathcal{M}^a}(x, \mathbf{A}) = \mathbf{dis}_{\mathcal{M}^a_{\mathbf{A}}}(x) \setminus \{x\}$. Moreover, $\mathbf{pre}_{\mathcal{M}^a, \prec}(x) \subseteq \mathbf{nd}_{\mathcal{M}^a}(x) \cup \{x\}$.

One can write $\mathbf{A} \setminus \{x\}$ as a function $g$ of all auxiliary ancestors $s_{\mathrm{aux}}$ of $\mathbf{A} \setminus \{x\}$ in $\mathcal{M}^a$ and of all noise ancestors $s_\epsilon$ of $\mathbf{A} \setminus \{x\}$ in $\mathcal{M}^a$, that is $\mathbf{A} \setminus \{x\} = g(s_{\mathrm{aux}}, s_\epsilon)$. Note that $x \notin s_{\mathrm{aux}}$ because $\mathbf{A} \setminus \{x\} \subseteq \mathbf{nd}_{\mathcal{M}^a}(x)$. Also note that $g$ is measurable because the $f_i$'s are measurable. Furthermore, because $\mathbf{A}$ is ancestral, $s_{\mathrm{aux}}$ and $s_\epsilon$ are contained in $\mathbf{A}$. Also, by assumption, $x \in \mathbf{A}$.

Now, suppose that

$$\mathbf{dis}_{\mathcal{M}^a_{\mathbf{A}}}(x) \not\perp\!\!\!\perp s_{\mathrm{aux}} \setminus \mathbf{dis}_{\mathcal{M}^a_{\mathbf{A}}}(x).$$

Then, by the extended Reichenbach common cause principle (Lemma 1), $\mathbf{dis}_{\mathcal{M}^a_{\mathbf{A}}}(x)$ and $s_{\mathrm{aux}} \setminus \mathbf{dis}_{\mathcal{M}^a_{\mathbf{A}}}(x)$ have a common ancestor in the full time graph. By construction of the augmented $(t_0, n)$-cutoff graph $\mathcal{M}^a$, there thus is a bidirected edge between some vertex in $\mathbf{dis}_{\mathcal{M}^a_{\mathbf{A}}}(x)$ and some vertex in $s_{\mathrm{aux}} \setminus \mathbf{dis}_{\mathcal{M}^a_{\mathbf{A}}}(x)$ in $\mathcal{M}^a$. Because $\mathbf{A}$ contains both $\mathbf{dis}_{\mathcal{M}^a_{\mathbf{A}}}(x)$ and

---

[4]See, e.g., Section 1.5 in Studený [2018] for this and other properties of conditional independence.

$s_{\text{aux}} \setminus \mathbf{dis}_{\mathcal{M}_{\boldsymbol{A}}^a}(x)$, there hence is a bidirected edge between the two sets in $\mathcal{M}_{\boldsymbol{A}}^a$. This fact, however, is a contradiction to the definition of districts. Thus,

$$\mathbf{dis}_{\mathcal{M}_{\boldsymbol{A}}^a}(x) \perp\!\!\!\perp s_{\text{aux}} \setminus \mathbf{dis}_{\mathcal{M}_{\boldsymbol{A}}^a}(x). \tag{8}$$

Therefore and because of Assumption (A3),

$$
\begin{aligned}
P_{\mathbf{dis}_{\mathcal{M}_{\boldsymbol{A}}^a}(x),\, s_{\text{aux}} \setminus \mathbf{dis}_{\mathcal{M}_{\boldsymbol{A}}^a}(x),\, s_\epsilon} &= P_{\mathbf{dis}_{\mathcal{M}_{\boldsymbol{A}}^a}(x),\, s_{\text{aux}} \setminus \mathbf{dis}_{\mathcal{M}_{\boldsymbol{A}}^a}(x)} \otimes P_{s_\epsilon} \\
&= P_{\mathbf{dis}_{\mathcal{M}_{\boldsymbol{A}}^a}(x)} \otimes P_{\boldsymbol{s}_{\text{aux}} \setminus \mathbf{dis}_{\mathcal{M}_{\boldsymbol{A}}^a}(x)} \otimes P_{\boldsymbol{s}_\epsilon} \\
&= P_{\mathbf{dis}_{\mathcal{M}_{\boldsymbol{A}}^a}(x)} \otimes P_{\boldsymbol{s}_{\text{aux}} \setminus \mathbf{dis}_{\mathcal{M}_{\boldsymbol{A}}^a}(x),\, s_\epsilon}.
\end{aligned}
$$

Thus,

$$\mathbf{dis}_{\mathcal{M}_{\boldsymbol{A}}^a}(x) \perp\!\!\!\perp (s_{\text{aux}} \setminus \mathbf{dis}_{\mathcal{M}_{\boldsymbol{A}}^a}(x), s_\epsilon).$$

By the weak union property of conditional independence and the fact that $x \in \mathbf{dis}_{\mathcal{M}_{\boldsymbol{A}}^a}(x)$ by definition,

$$x \perp\!\!\!\perp (s_{\text{aux}} \setminus \mathbf{dis}_{\mathcal{M}_{\boldsymbol{A}}^a}(x), s_\epsilon) \mid (\mathbf{dis}_{\mathcal{M}_{\boldsymbol{A}}^a}(x) \setminus \{x\}).$$

Also, from a standard property of conditional independence[5],

$$x \perp\!\!\!\perp (\mathbf{dis}_{\mathcal{M}_{\boldsymbol{A}}^a}(x) \setminus \{x\}) \mid (s_{\text{aux}} \setminus \mathbf{dis}_{\mathcal{M}_{\boldsymbol{A}}^a}(x), s_\epsilon, \mathbf{dis}_{\mathcal{M}_{\boldsymbol{A}}^a}(x) \setminus \{x\}).$$

Thus, by the contraction property of conditional independence

$$x \perp\!\!\!\perp (s_{\text{aux}}, s_\epsilon) \mid (\mathbf{dis}_{\mathcal{M}_{\boldsymbol{A}}^a}(x) \setminus \{x\})$$

and therefore,

$$x \perp\!\!\!\perp g(s_{\text{aux}}, s_\epsilon) \mid (\mathbf{dis}_{\mathcal{M}_{\boldsymbol{A}}^a}(x) \setminus \{x\}).$$

Thus,

$$x \perp\!\!\!\perp \boldsymbol{A} \setminus \{x\} \mid (\mathbf{dis}_{\mathcal{M}_{\boldsymbol{A}}^a}(x) \setminus \{x\}).$$

Hence, due to the decomposition property of conditional independence and because $\mathbf{mb}_{\mathcal{M}^a}(x, \boldsymbol{A}) = \mathbf{dis}_{\mathcal{M}_{\boldsymbol{A}}^a}(x) \setminus \{x\}$,

$$x \perp\!\!\!\perp \boldsymbol{A} \setminus (\mathbf{mb}_{\mathcal{M}^a}(x, \boldsymbol{A}) \cup \{x\}) \mid \mathbf{mb}_{\mathcal{M}^a}(x, \boldsymbol{A}).$$

**Remaining proof:**

Thus, we have shown the ordered local $m$-separation Markov property for the augmented $(t_0, n)$-cutoff graph and corresponding distribution of $\{(\boldsymbol{X}_t, \boldsymbol{\epsilon}_t)\}_{t \in \mathbb{Z}}$. As mentioned above, Theorem 2 in Richardson [2003] now yields the global $m$-separation Markov property for the augmented $(t_0, n)$-cutoff graph and corresponding distribution of $\{(\boldsymbol{X}_t, \boldsymbol{\epsilon}_t)\}_{t \in \mathbb{Z}}$.

Now, for any sets of non-noise vertices $\boldsymbol{S}_1$, $\boldsymbol{S}_2$ and $\boldsymbol{S}_3$, note that $\boldsymbol{S}_1$ and $\boldsymbol{S}_3$ are $m$-separated in $\mathcal{M}^a$ if and only if they are $m$-separated in $\mathcal{M}$ because the noise variables are at most adjacent to one non-noise variable and thus no new paths between $\boldsymbol{S}_1$ and $\boldsymbol{S}_2$ are created when considering $\mathcal{M}^a$. Therefore, from the global $m$-separation Markov property for the augmented graph follows the global $m$-separation Markov property for the not-augmented graph and corresponding distribution of $\{\boldsymbol{X}_t\}_{t \in \mathbb{Z}}$. $\qquad\square$

---

[5]This fact states that for random vectors $\boldsymbol{X}, \boldsymbol{Y}, \boldsymbol{Z}$, it holds that $\boldsymbol{X} \perp\!\!\!\perp \boldsymbol{Y} \mid (\boldsymbol{X}, \boldsymbol{Z})$, see, e.g., Definition 1.5.2 in Studený [2018] and the comment below this definition.

## C  FURTHER LEMMAS AND EXAMPLES

**Example 2** (Violation of Assumption (A2)). Consider the stochastic difference equations in equation (4) with $\phi = 1$ and the stochastic processes $\{\epsilon_t\}_{t\in\mathbb{Z}} = \{(\epsilon_t^1, \epsilon_t^2)\}_{t\in\mathbb{Z}}$ and $\{X_t\}_{t\in\mathbb{Z}} = \{(X_t^1, X_t^2)\}_{t\in\mathbb{Z}}$ where each $\epsilon_t$ is normally distributed with mean zero and identity covariance matrix, where $X_0^1$ is standard normally distributed and independent of $\{\epsilon_t\}_{t\in\mathbb{Z}}$, and where the remaining variables of $\{X_t\}_{t\in\mathbb{Z}}$ are generated by

$$X_{t+1}^1 = X_t^1 + X_t^2 + \epsilon_{t+1}^1 \quad \text{for } t > 0,$$
$$X_t^1 = X_{t+1}^1 - X_t^2 - \epsilon_{t+1}^1 \quad \text{for } t < 0, \text{ and}$$
$$X_t^2 = \epsilon_t^2 \text{ for all } t \in \mathbb{Z}.$$

Note that these equations are not the stochastic difference equations, these are just the equations that define $\{X_t\}_{t\in\mathbb{Z}}$.

The process $\{(X_t, \epsilon_t)\}_{t\in\mathbb{Z}}$ is a solution of the stochastic difference equations in (4): For $t > 0$, this claim follows by definition; for $t \leq 0$, this claim follows from rearranging $X_t^1 = X_{t+1}^1 - X_t^2 - \epsilon_t^1$. One can also see that $\{X_t\}_{t\in\mathbb{Z}}$ is not strictly stationary: The variance of $X_1^1$ equals 3, however, the variance of $X_0^1$ equals just 1 by definition. Also note that Assumption (A3) and the $d$-separation global Markov property are violated: We have

$$\begin{aligned}
\mathrm{Cov}(X_{-2}^1, X_{-1}^2) &= \mathrm{Cov}(X_{-2}^1, \epsilon_{-1}^2) \\
&= \mathrm{Cov}(X_{-1}^1 - X_{-2}^2 - \epsilon_{-1}^1, \epsilon_{-1}^2) \\
&= \mathrm{Cov}(X_{-1}^1 - \epsilon_{-2}^2 - \epsilon_{-1}^1, \epsilon_{-1}^2) \\
&= \mathrm{Cov}(X_{-1}^1, \epsilon_{-1}^2) \\
&= \mathrm{Cov}(X_0^1 - X_{-1}^2 - \epsilon_0^1, \epsilon_{-1}^2) \\
&= \mathrm{Cov}(X_0^1 - \epsilon_{-1}^2 - \epsilon_0^1, \epsilon_{-1}^2) \\
&= -\mathrm{Var}(\epsilon_{-1}^2) \\
&= -1
\end{aligned}$$

where we used that $\epsilon_{-2}^2, \epsilon_{-1}^1, \epsilon_{-1}^2, \epsilon_0^1$ and $X_0^1$ are independent of $\epsilon_{-1}^2$ by definition. However, in the full time graph $X_{-2}^1$ and $X_{-1}^2$ are $d$-separated given the empty set. The other assumptions are satisfied — the explanations are similar as for the violation of Assumption (A3) in the main paper.

So we have an example that violates Assumption (A1), but the real reason why the $d$-separation global Markov property is violated is the violation of Assumption (A3) because $X_t^2 = \epsilon_t^2$ for all $t \in \mathbb{Z}$ and thus a violation of Assumption (A3) directly carries over to a violation of the $d$-separation global Markov property.

**Lemma 4.** *Let $\phi \in \mathbb{R}$ with $|\phi| > 1$. Let $\epsilon_t := \{\epsilon_t^1, \epsilon_t^2\}_{t\in\mathbb{Z}}$ be a stochastic process satisfying (A0) and having finite second moments. Define the stochastic process $\{X_t\}_{t\in\mathbb{Z}} := \{X_t^1, X_t^2\}_{t\in\mathbb{Z}}$ for each $t \in \mathbb{Z}$ by*

$$X_t^1 = -\sum_{i=1}^{\infty} \phi^{-i}(\epsilon_{t+i}^1 + \epsilon_{t+i-1}^2) \quad \text{and}$$
$$X_t^2 = \epsilon_t^2.$$

*Then, $\{X_t\}_{t\in\mathbb{Z}}$ and $\{\epsilon_t\}_{t\in\mathbb{Z}}$ solve the VAR stochastic difference equations*

$$X_t^1 = \phi \cdot X_{t-1}^1 + X_{t-1}^2 + \epsilon_t^1 \quad \text{and}$$
$$X_t^2 = \epsilon_t^2$$

*$P$-almost surely for all $t \in \mathbb{Z}$. Moreover, define the time reversed processes $\{\tilde{X}_t\}_{t\in\mathbb{Z}}$ and $\{\tilde{\epsilon}_t\}_{t\in\mathbb{Z}}$ by $(\tilde{X}_t^1, \tilde{X}_t^2) = (X_{-t}^1, -X_{-t-1}^2/\phi)$ and $(\tilde{\epsilon}_t^1, \tilde{\epsilon}_t^2) = (-\epsilon_{-t+1}^1/\phi, -\epsilon_{-t-1}^2/\phi)$. Then, $\{\tilde{X}_t\}_{t\in\mathbb{Z}}$ and $\{\tilde{\epsilon}_t\}_{t\in\mathbb{Z}}$ satisfy the VAR stochastic difference equations*

$$\tilde{X}_t^1 = \frac{1}{\phi}\tilde{X}_{t-1}^1 + \tilde{X}_{t-1}^2 + \tilde{\epsilon}_t^1 \quad \text{and}$$
$$\tilde{X}_t^2 = \tilde{\epsilon}_t^2$$

*$P$-almost surely for all $t \in \mathbb{Z}$.*

*Proof.* First of all, note that $\{\phi^{-i}\}_{i\in\mathbb{N}_{\geq 1}}$ is absolutely summable and hence, $\sum_{i=1}^{\infty}\phi^{-i}(\epsilon_{t+i}^1 + \epsilon_{t+i-1}^2) :=$ $\lim_{N\to\infty}\sum_{i=1}^{N}\phi^{-i}(\epsilon_{t+i}^1 + \epsilon_{t+i-1}^2)$, interpreted as a limit in mean-square, exists and is uniquely defined up to $P$-nullsets [Lütkepohl, 2005, Section C.3]. Therefore, the below algebraic manipulations of this infinite sum such as multiplication by a scalar, shifting the summation index, and adding other random variables correspond to the same operations on the finite sums $\sum_{i=1}^{N}\phi^{-i}(\epsilon_{t+i}^1 + \epsilon_{t+i-1}^2)$ and then taking the limit.

**First part of the lemma:**

For an arbitrary but fixed $t \in \mathbb{Z}$, we $P$-almost surely have

$$\phi \cdot X_{t-1}^1 + X_{t-1}^2 + \epsilon_t^1 = \phi\cdot\left(-\sum_{i=1}^{\infty}\phi^{-i}(\epsilon_{t-1+i}^1 + \epsilon_{t-1+i-1}^2)\right) + \epsilon_{t-1}^2 + \epsilon_t^1$$

$$=\left(-\sum_{i=1}^{\infty}\phi^{-(i-1)}(\epsilon_{t+i-1}^1 + \epsilon_{t+(i-1)-1}^2)\right) + \epsilon_{t-1}^2 + \epsilon_t^1$$

$$=\left(-\sum_{i=0}^{\infty}\phi^{-i}(\epsilon_{t+i}^1 + \epsilon_{t+i-1}^2)\right) + \epsilon_{t-1}^2 + \epsilon_t^1$$

$$=\left(-\sum_{i=1}^{\infty}\phi^{-i}(\epsilon_{t+i}^1 + \epsilon_{t+i-1}^2)\right) - \epsilon_{t-1}^2 - \epsilon_t^1 + \epsilon_{t-1}^2 + \epsilon_t^1$$

$$= -\sum_{i=1}^{\infty}\phi^{-i}(\epsilon_{t+i}^1 + \epsilon_{t+i-1}^2)$$

$$= X_t^1.$$

Moreover, $X_t^2 = \epsilon_t^2$ trivially holds. Thus, we have finished the first part of the lemma.

**Second part of the lemma:** From the first part of the lemma, for an arbitrary but fixed $t \in \mathbb{Z}$, we $P$-almost surely have

$$X_{-t+1}^1 = \phi \cdot X_{-t}^1 + X_{-t}^2 + \epsilon_{-t+1}^1.$$

Rearranging this equation yields

$$X_{-t}^1 = \frac{1}{\phi}\left(X_{-t+1}^1 - X_{-t}^2 - \epsilon_{-t+1}^1\right).$$

Plugging in the definition of $\{\tilde{\boldsymbol{X}}_t\}_{t\in\mathbb{Z}}$ and $\{\tilde{\boldsymbol{\epsilon}}_t\}_{t\in\mathbb{Z}}$, we obtain

$$\tilde{X}_t^1 = \frac{1}{\phi}\tilde{X}_{t-1}^1 + \tilde{X}_{t-1}^2 + \tilde{\epsilon}_t^1.$$

Similarly, from the first part of the lemma, we have

$$X_{-t-1}^2 = \epsilon_{-t-1}^2.$$

Plugging in the definition of $\{\tilde{\boldsymbol{X}}_t\}_{t\in\mathbb{Z}}$ and $\{\tilde{\boldsymbol{\epsilon}}_t\}_{t\in\mathbb{Z}}$ yields

$$\tilde{X}_t^2 = \tilde{\epsilon}_t^2.$$

Thus, we have finished the second part of the lemma. $\qquad\square$

**Lemma 5.** *Let $\{\boldsymbol{X}_t\}_{t\in\mathbb{Z}}$ be a stochastic process such that each $\boldsymbol{X}_t$ is defined on the probability space $(\Omega, \mathcal{F}, P)$ and takes values in the measurable space $(S_{\boldsymbol{X}}, \Sigma_{\boldsymbol{X}})$. Furthermore, assume that $\{\boldsymbol{X}_t\}_{t\in\mathbb{Z}}$ is strictly stationary. Then for all $a \in \mathbb{Z}$ and $m \in \mathbb{N}_{\geq 0}$,*

$$\sup_{\substack{A\in\sigma(\boldsymbol{X}_t\colon t\leq a) \\ B\in\sigma(\boldsymbol{X}_t\colon t\geq a+m)}} \big|P(A\cap B) - P(A)P(B)\big| = \sup_{\substack{A\in\sigma(\boldsymbol{X}_t\colon t\leq 0) \\ B\in\sigma(\boldsymbol{X}_t\colon t\geq m)}} \big|P(A\cap B) - P(A)P(B)\big|. \tag{9}$$

*Proof.* Notational remark: In this proof, we always implicitly assume that time indices are in $\mathbb{Z}$.

Write

$$\alpha(m) := \sup_{\substack{A \in \sigma(\boldsymbol{X}_t:\, t \leq 0) \\ B \in \sigma(\boldsymbol{X}_t:\, t \geq m)}} \big| P(A \cap B) - P(A)P(B) \big|$$

Define

$$\mathcal{M}_{\leq a} := \{\boldsymbol{X}_t^{-1}(E):\ E \in \Sigma_{\boldsymbol{X}},\ t \leq a\}.$$

Recall that $\sigma(\boldsymbol{X}_t:\ t \leq a) = \sigma(\mathcal{M}_{\leq a})$. Furthermore, define

$$\mathcal{C}_{\leq a} := \left\{ \bigcup_{i \in I} \bigcap_{j \in J} \boldsymbol{X}_{t_{ij}}^{-1}(E_{ij}):\ I, J \text{ are finite index sets, and } \forall i, j: E_{ij} \in \Sigma_{\boldsymbol{X}},\ t_{ij} \leq a \right\}.$$

We now show that

$$\sigma(\mathcal{M}_{\leq a}) = \sigma(\mathcal{C}_{\leq a}).$$

Note that

$$\mathcal{M}_{\leq a} \subseteq \mathcal{C}_{\leq a},$$

so $\sigma(\mathcal{M}_{\leq a}) \subseteq \sigma(\mathcal{C}_{\leq a})$. Also note that every $\sigma$-algebra containing $\mathcal{M}_{\leq a}$ contains $\mathcal{C}_{\leq a}$, so

$$\mathcal{C}_{\leq a} \subseteq \sigma(\mathcal{M}_{\leq a}).$$

Therefore,

$$\sigma(\mathcal{C}_{\leq a}) \subseteq \sigma\big(\sigma(\mathcal{M}_{\leq a})\big) = \sigma(\mathcal{M}_{\leq a})$$

where the last equality follows because $\sigma(\mathcal{M}_{\leq a})$ is a $\sigma$-algebra and thus the smallest one containing itself. Hence, we conclude that $\sigma(\mathcal{C}_{\leq a}) = \sigma(\mathcal{M}_{\leq a})$.

We next show that $\mathcal{C}_{\leq a}$ is an algebra (see, e.g., §4 in Halmos [1974] for the notion of an algebra):

- $\Omega \in \mathcal{C}_{\leq a}$ because $S_{\boldsymbol{X}} \in \Sigma_{\boldsymbol{X}}$ and for all $t \leq a$, it holds that $\boldsymbol{X}_t^{-1}(S_{\boldsymbol{X}}) = \Omega$.
- If $A \in \mathcal{C}_{\leq a}$, then $A^c \in \mathcal{C}_{\leq a}$: Write $A := \bigcup_{i \in I} \bigcap_{j \in J} \boldsymbol{X}_{t_{ij}}^{-1}(E_{ij})$. Then,

$$\begin{aligned}
A^c &= \left( \bigcup_{i \in I} \bigcap_{j \in J} \boldsymbol{X}_{t_{ij}}^{-1}(E_{ij}) \right)^c \\
&= \bigcap_{i \in I} \left( \bigcap_{j \in J} \boldsymbol{X}_{t_{ij}}^{-1}(E_{ij}) \right)^c \\
&= \bigcap_{i \in I} \bigcup_{j \in J} \left( \boldsymbol{X}_{t_{ij}}^{-1}(E_{ij}) \right)^c \\
&= \bigcap_{i \in I} \bigcup_{j \in J} \boldsymbol{X}_{t_{ij}}^{-1}(E_{ij}^c).
\end{aligned}$$

  The last line is again a finite union of finite intersections of preimages, just apply the distributive law for unions and intersections (however, one cannot simply interchange $\bigcap_{i \in I}$ and $\bigcup_{j \in J}$). So because every $E_{ij}^c \in \Sigma_{\boldsymbol{X}}$ since $\Sigma_{\boldsymbol{X}}$ is a $\sigma$-algebra, $A^c \in \mathcal{C}_{\leq a}$.

- If $A, B \in \mathcal{C}_{\leq a}$, then $A \cup B \in \mathcal{C}_{\leq a}$ : This result follows immediately since the union of two finite unions of finite intersections of preimages is again a finite union of finite intersections of preimages, just some relabelling is necessary.

Therefore, $\mathcal{C}_{\leq a}$ is an algebra and we have that $\sigma(\boldsymbol{X}_t : \ t \leq a) = \sigma(\mathcal{C}_{\leq a})$.

An analogous result holds for $\sigma(\boldsymbol{X}_t : \ t \geq a + m)$ and

$$\mathcal{C}_{\geq a+m} := \left\{ \bigcup_{i \in I} \bigcap_{j \in J} \boldsymbol{X}_{t_{ij}}^{-1}(F_{ij}) : \ I, J \text{ are finite index sets, and } \forall i, j : F_{ij} \in \Sigma_{\boldsymbol{X}}, \ t_{ij} \geq a + m \right\},$$

so $\sigma(\boldsymbol{X}_t : \ t \geq a + m) = \sigma(\mathcal{C}_{\geq a+m})$ and $\mathcal{C}_{\geq a+m}$ is an algebra.

Now, for all $A = \bigcup_{i \in I_1} \bigcap_{j \in J_1} \boldsymbol{X}_{t_{ij}}^{-1}(E_{ij}) \in \mathcal{C}_{\leq a}$ and for all $B = \bigcup_{i \in I_2} \bigcap_{j \in J_2} \boldsymbol{X}_{t_{ij}}^{-1}(F_{ij}) \in \mathcal{C}_{\geq a+m}$, define $A' = \bigcup_{i \in I_1} \bigcap_{j \in J_1} \boldsymbol{X}_{t_{ij}-a}^{-1}(E_{ij})$ and $B' = \bigcup_{i \in I_2} \bigcap_{j \in J_2} \boldsymbol{X}_{t_{ij}-a}^{-1}(F_{ij})$. Clearly, $A' \in \sigma(\boldsymbol{X}_t : \ t \leq 0)$ and $B' \in \sigma(\boldsymbol{X}_t : \ t \geq m)$. By strict stationarity and the inclusion-exclusion formula,

$$P(A \cap B) - P(A)P(B) = P(A' \cap B') - P(A')P(B'),$$

Thus,

$$|P(A \cap B) - P(A)P(B)| = |P(A' \cap B') - P(A')P(B')| \leq \alpha(m).$$

For general sets $A \in \sigma(\boldsymbol{X}_t : \ t \leq a)$ and $B \in \sigma(\boldsymbol{X}_t : \ t \geq a + m)$, we use the following result from measure theory (see, e.g., Theorem D in §13 in Halmos [1974]) that relies on the generating set of a $\sigma$-algebra being an algebra: For all $A \in \sigma(\boldsymbol{X}_t : \ t \leq a)$, for all $B \in \sigma(\boldsymbol{X}_t : \ t \geq a + m)$ and for all $\varepsilon > 0$, one can find $\hat{A} \in \mathcal{C}_{\leq a}$ and $\hat{B} \in \mathcal{C}_{\geq a+m}$ such that

$$P\left( (A \setminus \hat{A}) \cup (\hat{A} \setminus A) \right) \leq \tilde{\varepsilon} \quad \text{and}$$

$$P\left( (B \setminus \hat{B}) \cup (\hat{B} \setminus B) \right) \leq \tilde{\varepsilon}, \tag{10}$$

where $\tilde{\varepsilon} := -2 + \sqrt{4 + \varepsilon}$; note that $\tilde{\varepsilon}^2 + 4\tilde{\varepsilon} = \varepsilon$. Also note that (10) implies

$$P\left( \left[ (A \cap B) \setminus (\hat{A} \cap \hat{B}) \right] \cup \left[ (\hat{A} \cap \hat{B}) \setminus (A \cap B) \right] \right)$$

$$\leq P\left( (A \setminus \hat{A}) \cup (\hat{A} \setminus A) \cup (B \setminus \hat{B}) \cup (\hat{B} \setminus B) \right)$$

$$\leq P\left( (A \setminus \hat{A}) \cup (\hat{A} \setminus A) \right) + P\left( (B \setminus \hat{B}) \cup (\hat{B} \setminus B) \right)$$

$$\leq \tilde{\varepsilon} + \tilde{\varepsilon}$$

$$= 2\tilde{\varepsilon}.$$

Here, the first inequality follows from

$$\left[ (A \cap B) \setminus (\hat{A} \cap \hat{B}) \right] \cup \left[ (\hat{A} \cap \hat{B}) \setminus (A \cap B) \right] \subseteq (A \setminus \hat{A}) \cup (\hat{A} \setminus A) \cup (B \setminus \hat{B}) \cup (\hat{B} \setminus B). \tag{11}$$

The subset relation in (11) is true due to following argumentation: Let

$$\omega \in \left[ (A \cap B) \setminus (\hat{A} \cap \hat{B}) \right] \cup \left[ (\hat{A} \cap \hat{B}) \setminus (A \cap B) \right].$$

Without loss of generality, assume that

$$\omega \in (A \cap B) \setminus (\hat{A} \cap \hat{B}).$$

Then,

$$\omega \in A \setminus (\hat{A} \cap \hat{B}).$$

Now, suppose that

$$\omega \notin A \setminus \hat{A}.$$

Then, $\omega \in A$ and $\omega \in \hat{A}$ but $\omega \notin \hat{B}$. Because $\omega \in (A \cap B) \setminus (\hat{A} \cap \hat{B})$, it follows that $\omega \in B$ and hence, $\omega \in B \setminus \hat{B}$. Arguing analogously for the other cases, we conclude that the subset-relation in (11) is true.

Combining the previous approximations of the probability measures, we now have

$$
\begin{aligned}
& P(\hat{A}) - \tilde{\varepsilon} \\
&= P(A \cap \hat{A}) + P(\hat{A} \setminus A) - \tilde{\varepsilon} \\
&\leq P(A) + P(\hat{A} \setminus A) - \tilde{\varepsilon} \\
&\leq P(A) + P\Big( (A \setminus \hat{A}) \cup (\hat{A} \setminus A) \Big) - \tilde{\varepsilon} \\
&\leq P(A) \\
&= P(A \cap \hat{A}) + P(A \setminus \hat{A}) \\
&\leq P(\hat{A}) + P(A \setminus \hat{A}) \\
&\leq P(\hat{A}) + P\Big( (A \setminus \hat{A}) \cup (\hat{A} \setminus A) \Big) \\
&\leq P(\hat{A}) + \tilde{\varepsilon}.
\end{aligned}
$$

Similarly, $P(\hat{B}) - \tilde{\varepsilon} \leq P(B) \leq P(\hat{B}) + \tilde{\varepsilon}$ and $P(\hat{A} \cap \hat{B}) - 2\tilde{\varepsilon} \leq P(A \cap B) \leq P(\hat{A} \cap \hat{B}) + 2\tilde{\varepsilon}$. Thus,

$$
\begin{aligned}
& P(\hat{A} \cap \hat{B}) - P(\hat{A})P(\hat{B}) - \varepsilon \\
&= P(\hat{A} \cap \hat{B}) - P(\hat{A})P(\hat{B}) - 4\tilde{\varepsilon} - \tilde{\varepsilon}^2 \\
&\leq P(\hat{A} \cap \hat{B}) - P(\hat{A})P(\hat{B}) - 2\tilde{\varepsilon} - \Big( P(\hat{A}) + P(\hat{B}) \Big) \cdot \tilde{\varepsilon} - \tilde{\varepsilon}^2 \\
&= \Big( P(\hat{A} \cap \hat{B}) - 2\tilde{\varepsilon} \Big) - \Big( P(\hat{A}) + \tilde{\varepsilon} \Big) \cdot \Big( P(\hat{B}) + \tilde{\varepsilon} \Big) \\
&\leq P(A \cap B) - P(A)P(B) \\
&\leq \Big( P(\hat{A} \cap \hat{B}) + 2\tilde{\varepsilon} \Big) - \Big( P(\hat{A}) - \tilde{\varepsilon} \Big) \cdot \Big( P(\hat{B}) - \tilde{\varepsilon} \Big) \\
&= P(\hat{A} \cap \hat{B}) - P(\hat{A})P(\hat{B}) + 2\tilde{\varepsilon} + \Big( P(\hat{A}) + P(\hat{B}) \Big) \cdot \tilde{\varepsilon} - \tilde{\varepsilon}^2 \\
&\leq P(\hat{A} \cap \hat{B}) - P(\hat{A})P(\hat{B}) + 4\tilde{\varepsilon} - \tilde{\varepsilon}^2 \\
&\leq P(\hat{A} \cap \hat{B}) - P(\hat{A})P(\hat{B}) + 4\tilde{\varepsilon} + \tilde{\varepsilon}^2 \\
&= P(\hat{A} \cap \hat{B}) - P(\hat{A})P(\hat{B}) + \varepsilon
\end{aligned}
$$

Therefore,

$$
\begin{aligned}
|P(A \cap B) - P(A)P(B)| &\leq \max\Big\{ \big| P(\hat{A} \cap \hat{B}) - P(\hat{A})P(\hat{B}) - \varepsilon \big|, \big| P(\hat{A} \cap \hat{B}) - P(\hat{A})P(\hat{B}) + \varepsilon \big| \Big\} \\
&\leq \max\Big\{ \big| P(\hat{A} \cap \hat{B}) - P(\hat{A})P(\hat{B}) \big| + \varepsilon, \big| P(\hat{A} \cap \hat{B}) - P(\hat{A})P(\hat{B}) \big| + \varepsilon \Big\} \\
&= \big| P(\hat{A} \cap \hat{B}) - P(\hat{A})P(\hat{B}) \big| + \varepsilon \\
&\leq \alpha(m) + \varepsilon.
\end{aligned}
$$

Because $\varepsilon > 0$ was arbitrary, $|P(A \cap B) - P(A)P(B)| \leq \alpha(m)$. Therefore,

$$\sup_{\substack{A \in \sigma(\boldsymbol{X}_t \colon t \leq a) \\ B \in \sigma(\boldsymbol{X}_t \colon t \geq a+m)}} \big| P(A \cap B) - P(A)P(B) \big| \leq \alpha(m).$$

Arguing analogously the other way round, we conclude that equation (9) is true. $\qquad \square$

**Lemma 6.** *Let $\{\boldsymbol{X}_t\}_{t\in\mathbb{Z}}$ be a stochastic process such that each $\boldsymbol{X}_t$ is defined on the probability space $(\Omega, \mathcal{F}, P)$ and takes values in the measurable space $(S_{\boldsymbol{X}}, \Sigma_{\boldsymbol{X}})$. Furthermore, assume that $\{\boldsymbol{X}_t\}_{t\in\mathbb{Z}}$ is strictly stationary and $\alpha$-strongly mixing. Then, the time-reversed process $\{\tilde{\boldsymbol{X}}_t\}_{t\in\mathbb{Z}}$ defined by $\tilde{\boldsymbol{X}}_t := \boldsymbol{X}_{-t}$ is also $\alpha$-strongly mixing.*

*Proof.* Notational remark: In this proof, we always implicitly assume that time indices are in $\mathbb{Z}$.

For arbitrary but fixed $m \in \mathbb{N}_{\geq 0}$,

$$
\begin{aligned}
\alpha_{\tilde{\boldsymbol{X}}}(m) &:= \sup_{\substack{A \in \sigma(\tilde{\boldsymbol{X}}_t:\, t\leq 0) \\ B \in \sigma(\tilde{\boldsymbol{X}}_t:\, t\geq m)}} \big|P(A\cap B) - P(A)P(B)\big| \\
&= \sup_{\substack{A \in \sigma(\boldsymbol{X}_{-t}:\, t\leq 0) \\ B \in \sigma(\boldsymbol{X}_{-t}:\, t\geq m)}} \big|P(A\cap B) - P(A)P(B)\big| \\
&= \sup_{\substack{A \in \sigma(\boldsymbol{X}_{-t}:\, -t\geq 0) \\ B \in \sigma(\boldsymbol{X}_{-t}:\, -t\leq -m)}} \big|P(A\cap B) - P(A)P(B)\big| \\
&= \sup_{\substack{A \in \sigma(\boldsymbol{X}_t:\, t\geq 0) \\ B \in \sigma(\boldsymbol{X}_t:\, t\leq -m)}} \big|P(A\cap B) - P(A)P(B)\big| \\
&\overset{(*)}{=} \sup_{\substack{A \in \sigma(\boldsymbol{X}_t:\, t\geq m) \\ B \in \sigma(\boldsymbol{X}_t:\, t\leq 0)}} \big|P(A\cap B) - P(A)P(B)\big| \\
&\overset{(**)}{=} \sup_{\substack{A \in \sigma(\boldsymbol{X}_t:\, t\leq 0) \\ B \in \sigma(\boldsymbol{X}_t:\, t\geq m)}} \big|P(A\cap B) - P(A)P(B)\big| \\
&=: \alpha_{\boldsymbol{X}}(m)
\end{aligned}
$$

Here, $(*)$ follows from Lemma 5 and $(**)$ follows from relabelling. Thus, we obtain the statement that we wanted to prove. $\qquad\square$

**Lemma 7.** *Let $\{\boldsymbol{X}_t\}_{t\in\mathbb{Z}} = \{(X_t^1, X_t^2)\}_{t\in\mathbb{Z}}$ be a stochastic process such that each $X_t^1$ and each $X_t^2$ is defined on the probability space $(\Omega, \mathcal{F}, P)$ and takes values in the measurable space $(\mathcal{X}_1, \Sigma_{X_1})$ respectively $(\mathcal{X}_2, \Sigma_{X_2})$. Define a new stochastic process $\{\tilde{\boldsymbol{X}}_t\}_{t\in\mathbb{Z}}$ by $\tilde{\boldsymbol{X}}_t = (\tilde{\boldsymbol{X}}_t^1, \tilde{\boldsymbol{X}}_t^2) := (\boldsymbol{X}_{t-k}^1, \boldsymbol{X}_t^2)$ for some fixed $k \geq 0$. If $\{\boldsymbol{X}_t\}_{t\in\mathbb{Z}}$ is $\alpha$-strongly mixing, then $\{\tilde{\boldsymbol{X}}_t\}_{t\in\mathbb{Z}}$ is $\alpha$-strongly mixing.*

*Proof.* Remark: In this proof, we always implicitly assume that time indices are in $\mathbb{Z}$.

Let $m \in \mathbb{N}_{\geq k}$ be arbitrary but fixed. Write

$$
\alpha_{\boldsymbol{X}}(m) := \sup_{\substack{A \in \sigma(\boldsymbol{X}_t:\, t\leq 0) \\ B \in \sigma(\boldsymbol{X}_t:\, t\geq m)}} \big|P(A\cap B) - P(A)P(B)\big|.
$$

Define

$$
\mathcal{S}_{\leq 0} := \{\tilde{\boldsymbol{X}}_t^{-1}(E):\ E \in \Sigma_1^X \otimes \Sigma_2^X,\ t \leq 0\}.
$$

Recall that $\sigma(\tilde{\boldsymbol{X}}_t:\ t \leq 0) = \sigma(\mathcal{S}_{\leq 0})$. Furthermore, define

$$
\mathcal{M}_{\leq 0} := \{\tilde{\boldsymbol{X}}_t^{-1}(E^1 \times E^2):\ E^1 \in \Sigma_1^X,\ E^2 \in \Sigma_2^X,\ t \leq 0\}.
$$

and

$$
\mathcal{C}_{\leq 0} := \left\{ \bigcup_{i\in I}\bigcap_{j\in J}(\tilde{X}_{t_{ij}}^{l_{ij}})^{-1}(E_{ij}):\ I, J \text{ are finite index sets, and } \forall i,j: t_{ij} \leq 0,\ l_{ij} \in \{1,2\},\ E_{ij} \in \Sigma_{l_{ij}}^X; \right\}.
$$

We first show that

$$
\sigma(\mathcal{M}_{\leq 0}) = \sigma(\mathcal{C}_{\leq 0}).
$$

First, note that for all $E^1 \in \Sigma_1^X$ and for all $E^2 \in \Sigma_2^X$,

$$\tilde{\boldsymbol{X}}_t^{-1}(E^1 \times E^2) = \left(\tilde{X}_t^1\right)^{-1}(E^1) \cap \left(\tilde{X}_t^2\right)^{-1}(E^2).$$

Thus,

$$\mathcal{M}_{\leq 0} \subseteq \mathcal{C}_{\leq 0}$$

and hence,

$$\sigma(\mathcal{M}_{\leq 0}) \subseteq \sigma(\mathcal{C}_{\leq 0}).$$

Vice versa, note that every $\sigma$-algebra containing $\mathcal{M}_{\leq 0}$ contains

$$\tilde{\boldsymbol{X}}_t^{-1}(E^1 \times \mathcal{X}_2) = \left(\tilde{X}_t^1\right)^{-1}(E^1) \cap \left(\tilde{X}_t^2\right)^{-1}(\mathcal{X}_2) = \left(\tilde{X}_t^1\right)^{-1}(E^1) \cap \Omega = \left(\tilde{X}_t^1\right)^{-1}(E^1)$$

for every $E_1 \in \Sigma_1^X$. Similarly, every such $\sigma$-algebra contains

$$\left(\tilde{X}_t^2\right)^{-1}(E^2)$$

for every $E_2 \in \Sigma_2^X$. Thus, every $\sigma$-algebra containing $\mathcal{M}_{\leq 0}$ contains $\mathcal{C}_{\leq 0}$ and hence,

$$\sigma(\mathcal{C}_{\leq 0}) \subseteq \sigma(\sigma(\mathcal{M}_{\leq 0})) = \sigma(\mathcal{M}_{\leq 0}),$$

where the last equality follows because $\sigma(\mathcal{M}_{\leq 0})$ is a $\sigma$-algebra and thus the smallest one containing itself. Therefore, we have shown $\sigma(\mathcal{M}_{\leq 0}) = \sigma(\mathcal{C}_{\leq 0})$.

We next show that

$$\sigma(\mathcal{S}_{\leq 0}) = \sigma(\mathcal{M}_{\leq 0}).$$

First, we (trivially) have that

$$\mathcal{M}_{\leq 0} \subseteq \mathcal{S}_{\leq 0},$$

so

$$\sigma(\mathcal{M}_{\leq 0}) \subseteq \sigma(\mathcal{S}_{\leq 0}).$$

Vice versa, we argue as follows: Note that for all $E := E^1 \times E^2$ with $E^1 \in \Sigma_1^X$ and $E^2 \in \Sigma_2^X$ and for all $t \leq 0$, it (trivially) holds that

$$\tilde{\boldsymbol{X}}_t^{-1}(E) \in \sigma\left(\{\tilde{\boldsymbol{X}}_t^{-1}(E'^1 \times E'^2) : E'^1 \in \Sigma_1^X, \; E'^2 \in \Sigma_2^X\}\right).$$

Now, recall that by definition of $\Sigma_1^X \otimes \Sigma_2^X$,

$$\sigma\left(\{E^1 \times E^2 : E'^1 \in \Sigma_1^X, \; E'^2 \in \Sigma_2^X\}\right) = \Sigma_1^X \otimes \Sigma_2^X.$$

From a standard result in measure theory (see, e.g., Theorem 1.3.1 in Durrett [2019]), it then follows that

$$\tilde{\boldsymbol{X}}_t^{-1}(E) \in \sigma\left(\{\tilde{\boldsymbol{X}}_t^{-1}(E'^1 \times E'^2) : E'^1 \in \Sigma_1^X, \; E'^2 \in \Sigma_2^X\}\right)$$

for all $E \in \Sigma_1^X \otimes \Sigma_2^X$ and for all $t \leq 0$. Thus,

$$\sigma(\tilde{\boldsymbol{X}}_t) \subseteq \sigma\left(\{\tilde{\boldsymbol{X}}_t^{-1}(E'^1 \times E'^2) : E'^1 \in \Sigma_1^X, \; E'^2 \in \Sigma_2^X\}\right).$$

Vice versa, it is immediately clear by definition that

$$\sigma(\tilde{\boldsymbol{X}}_t) \supseteq \sigma\left(\{\tilde{\boldsymbol{X}}_t^{-1}(E'^1 \times E'^2) : E'^1 \in \Sigma_1^X, \; E'^2 \in \Sigma_2^X\}\right),$$

so, we conclude that

$$\sigma(\tilde{\boldsymbol{X}}_t) = \sigma\big(\{\tilde{\boldsymbol{X}}_t^{-1}(E'^1 \times E'^2) : \ E'^1 \in \Sigma_1^X, \ E'^2 \in \Sigma_2^X\}\big).$$

Therefore,

$$\sigma(\mathcal{S}_{\leq 0}) = \sigma\bigg(\bigcup_{t \leq 0} \sigma(\tilde{\boldsymbol{X}}_t)\bigg) = \sigma\bigg(\bigcup_{t \leq 0} \sigma(\{\tilde{\boldsymbol{X}}_t^{-1}(E'^1 \times E'^2) : \ E'^1 \in \Sigma_1^X, \ E'^2 \in \Sigma_2^X\})\bigg).$$

Now, by definition, for all $t \leq 0$,

$$\{\tilde{\boldsymbol{X}}_t^{-1}(E'^1 \times E'^2) : \ E'^1 \in \Sigma_1^X, \ E'^2 \in \Sigma_2^X\} \subseteq \sigma(\mathcal{M}_{\leq 0}).$$

So, for all $t \leq 0$,

$$\sigma(\{\tilde{\boldsymbol{X}}_t^{-1}(E'^1 \times E'^2) : \ E'^1 \in \Sigma_1^X, \ E'^2 \in \Sigma_2^X\}) \subseteq \sigma(\sigma(\mathcal{M}_{\leq 0})) = \sigma(\mathcal{M}_{\leq 0}),$$

where the last equality follows because $\sigma(\mathcal{M}_{\leq 0})$ is a $\sigma$-algebra and thus the smallest one containing itself. Hence,

$$\bigcup_{t \leq 0} \sigma(\{\tilde{\boldsymbol{X}}_t^{-1}(E'^1 \times E'^2) : \ E'^1 \in \Sigma_1^X, \ E'^2 \in \Sigma_2^X\}) \subseteq \sigma(\mathcal{M}_{\leq 0})$$

and thus,

$$\sigma(\mathcal{S}_{\leq 0}) = \sigma\bigg(\bigcup_{t \leq 0} \sigma(\{\tilde{\boldsymbol{X}}_t^{-1}(E'^1 \times E'^2) : \ E'^1 \in \Sigma_1^X, \ E'^2 \in \Sigma_2^X\})\bigg) \subseteq \sigma(\sigma(\mathcal{M}_{\leq 0})) = \sigma(\mathcal{M}_{\leq 0}),$$

where the last equality follows because $\sigma(\mathcal{M}_{\leq 0})$ is a $\sigma$-algebra and thus the smallest one containing itself. Therefore, $\sigma(\mathcal{C}_{\leq 0}) = \sigma(\mathcal{M}_{\leq 0}) = \sigma(\mathcal{S}_{\leq 0}) = \sigma(\tilde{\boldsymbol{X}}_t : \ t \leq 0)$.

By a similar argument as in the proof of Lemma 6, one can show that $\mathcal{C}_{\leq 0}$ is an algebra.

In an analogous fashion, one can also show that

$$\mathcal{C}_{\geq m} := \bigg\{ \bigcup_{i \in I} \bigcap_{j \in J} (\tilde{X}_{t_{ij}}^{l_{ij}})^{-1}(F_{ij}) : \ I, J \text{ are finite index sets, and } \forall i, j : t_{ij} \geq m, \ l_{ij} \in \{1, 2\}, \ F_{ij} \in \Sigma_{l_{ij}}^X \bigg\},$$

is an algebra with $\sigma(\mathcal{C}_{\geq m}) = \sigma(\tilde{\boldsymbol{X}}_t : \ t \geq m)$.

Now, for all sets $A = \bigcup_{i \in I_1} \bigcap_{j \in J_1} (\tilde{X}_{t_{ij}}^{l_{ij}})^{-1}(E_{ij}) \in \mathcal{C}_{\leq 0}$ and for all sets $B = \bigcup_{i \in I_2} \bigcap_{j \in J_2} (\tilde{X}_{t_{ij}}^{l_{ij}})^{-1}(F_{ij}) \in \mathcal{C}_{\geq m}$, we can by definition write $A = \bigcup_{i \in I_1} \bigcap_{j \in J_1} (X_{t_{ij} - k \cdot 1_{l_{ij} = 1}}^{l_{ij}})^{-1}(E_{ij})$ and $B = \bigcup_{i \in I_2} \bigcap_{j \in J_2} (X_{t_{ij} - k \cdot 1_{l_{ij} = 1}}^{l_{ij}})^{-1}(F_{ij})$. Thus, $A \in \sigma(X_t : \ t \leq 0)$ and $B \in \sigma(X_t : \ t \geq m - k)$. Therefore,

$$|P(A \cap B) - P(A)P(B)| \leq \alpha_{\boldsymbol{X}}(m - k).$$

For general sets $A \in \sigma(\tilde{\boldsymbol{X}}_t : \ t \leq 0)$ and $B \in \sigma(\tilde{\boldsymbol{X}}_t : \ t \geq m)$, one can do the exact same $\varepsilon$-approximation argument as in the proof of Lemma 6 because $\mathcal{C}_{\leq 0}$ and $\mathcal{C}_{\geq m}$ are algebras.

Therefore, we conclude that

$$\alpha_{\tilde{\boldsymbol{X}}}(m) := \sup_{\substack{A \in \sigma(\boldsymbol{X}_t : \ t \leq 0) \\ B \in \sigma(\boldsymbol{X}_t : \ t \geq m)}} |P(A \cap B) - P(A)P(B)| \leq \alpha_{\boldsymbol{X}}(m - k).$$

Because $\{\boldsymbol{X}_t\}_{t \in \mathbb{Z}}$ is $\alpha$-strongly mixing, and hence, $\alpha_{\boldsymbol{X}}(m - k) \to 0$ for $m \to \infty$, we obtain, $\alpha_{\tilde{\boldsymbol{X}}}(m) \to 0$ for $m \to \infty$. $\quad\square$

**Lemma 8.** *Suppose that a sequence of $\mathbb{R}^{d_1}$-valued random variables $\{\boldsymbol{Z}_n\}_{n \in \mathbb{N}_{\geq 1}}$ converges in probability to an $\mathbb{R}^{d_1}$-valued random variable $\boldsymbol{Z}$, in short, $\boldsymbol{Z}_n \xrightarrow{p} \boldsymbol{Z}$. Moreover, let $\boldsymbol{S}$ be an $\mathbb{R}^{d_2}$-valued random variable and suppose that for all $n \in \mathbb{N}_{\geq 1}$ it holds that $\boldsymbol{Z}_n \perp\!\!\!\perp \boldsymbol{S}$. Then, $\boldsymbol{Z} \perp\!\!\!\perp \boldsymbol{S}$.*

*Proof.* Notational remark: For vectors $\boldsymbol{a} \in \mathbb{R}^{d_1}$ and $\boldsymbol{b} \in \mathbb{R}^{d_2}$, we write $(\boldsymbol{a}, \boldsymbol{b}) \in \mathbb{R}^{d_1+d_2}$ to denote the vector whose first components are the components of $\boldsymbol{a}$ and whose second components are the components of $\boldsymbol{b}$. We write $\boldsymbol{a} \cdot \boldsymbol{b}$ to denote the standard scalar product between $\boldsymbol{a}$ and $\boldsymbol{b}$.

Note that $\boldsymbol{Z}_n \xrightarrow{P} \boldsymbol{Z}$ and $\boldsymbol{S} \xrightarrow{P} \boldsymbol{S}$ implies that $(\boldsymbol{Z}_n, \boldsymbol{S}) \xrightarrow{P} (\boldsymbol{Z}, \boldsymbol{S})$. Thus, $(\boldsymbol{Z}_n, \boldsymbol{S}) \xrightarrow{d} (\boldsymbol{Z}, \boldsymbol{S})$. By Levy's continuity theorem [Durrett, 2019, Theorem 3.3.17],

$$\lim_{n \to \infty} \mathbb{E}[e^{i(\boldsymbol{t}_1, \boldsymbol{t}_2) \cdot (\boldsymbol{Z}_n, \boldsymbol{S})}] = \mathbb{E}[e^{i(\boldsymbol{t}_1, \boldsymbol{t}_2) \cdot (\boldsymbol{Z}, \boldsymbol{S})}]$$

for all $\boldsymbol{t}_1 \in \mathbb{R}^{d_1}$ and for all $\boldsymbol{t}_2 \in \mathbb{R}^{d_2}$. Also, by Levy's continuity theorem,

$$\lim_{n \to \infty} \mathbb{E}[e^{i\boldsymbol{t}_1 \cdot \boldsymbol{Z}_n}] = \mathbb{E}[e^{i\boldsymbol{t}_1 \cdot \boldsymbol{Z}}]$$

for all $\boldsymbol{t}_1 \in \mathbb{R}^{d_1}$. Now, because $\boldsymbol{Z}_n \perp\!\!\!\perp \boldsymbol{S}$ for all $n \in \mathbb{N}$, we have

$$\mathbb{E}[e^{i(\boldsymbol{t}_1, \boldsymbol{t}_2) \cdot (\boldsymbol{Z}_n, \boldsymbol{S})}] = \mathbb{E}[e^{i\boldsymbol{t}_1 \cdot \boldsymbol{Z}_n}]\mathbb{E}[e^{i\boldsymbol{t}_2 \cdot \boldsymbol{S}}]$$

for all $\boldsymbol{t}_1 \in \mathbb{R}^{d_1}$ and for all $\boldsymbol{t}_2 \in \mathbb{R}^{d_2}$.

Thus, for all $\boldsymbol{t}_1 \in \mathbb{R}^{d_1}$ and for all $\boldsymbol{t}_2 \in \mathbb{R}^{d_2}$,

$$\begin{aligned}
\mathbb{E}[e^{i(\boldsymbol{t}_1, \boldsymbol{t}_2) \cdot (\boldsymbol{Z}, \boldsymbol{S})}] &= \lim_{n \to \infty} \mathbb{E}[e^{i(\boldsymbol{t}_1, \boldsymbol{t}_2) \cdot (\boldsymbol{Z}_n, \boldsymbol{S})}] \\
&= \lim_{n \to \infty} \mathbb{E}[e^{i\boldsymbol{t}_1 \cdot \boldsymbol{Z}_n}]\mathbb{E}[e^{i\boldsymbol{t}_2 \cdot \boldsymbol{S}}] \\
&= \mathbb{E}[e^{i\boldsymbol{t}_2 \cdot \boldsymbol{S}}] \lim_{n \to \infty} \mathbb{E}[e^{i\boldsymbol{t}_1 \cdot \boldsymbol{Z}_n}] \\
&= \mathbb{E}[e^{i\boldsymbol{t}_1 \cdot \boldsymbol{Z}}]\mathbb{E}[e^{i\boldsymbol{t}_2 \cdot \boldsymbol{S}}].
\end{aligned}$$

So $(\boldsymbol{Z}, \boldsymbol{S}) \sim P_{\boldsymbol{Z}, \boldsymbol{S}}$ has characteristic function $\mathbb{E}[e^{i\boldsymbol{t}_1 \cdot \boldsymbol{Z}}]\mathbb{E}[e^{i\boldsymbol{t}_2 \cdot \boldsymbol{S}}]$. The random vector $(\tilde{\boldsymbol{Z}}, \tilde{\boldsymbol{S}}) \sim P_{\boldsymbol{Z}} \otimes P_{\boldsymbol{S}}$ also has characteristic function $\mathbb{E}[e^{i\boldsymbol{t}_1 \cdot \boldsymbol{Z}}]\mathbb{E}[e^{i\boldsymbol{t}_2 \cdot \boldsymbol{S}}]$. Because equality of characteristic functions implies equality of the corresponding distributions [Durrett, 2019, Theorem 3.3.11], we conclude that $P_{\boldsymbol{Z}, \boldsymbol{S}} = P_{\boldsymbol{Z}} \otimes P_{\boldsymbol{S}}$. Hence, $\boldsymbol{Z} \perp\!\!\!\perp \boldsymbol{S}$ $\qquad \square$

**Lemma 9.** *Suppose that a sequence of $\mathbb{R}^d$-valued random variables $\{\boldsymbol{X}_n\}_{n \in \mathbb{N}}$ converges in distribution to an $\mathbb{R}^d$-valued random variable $\boldsymbol{X}$, in short $\boldsymbol{X}_n \xrightarrow{d} \boldsymbol{X}$. Similarly, suppose that a sequence of $\mathbb{R}^d$-valued random variables $\{\boldsymbol{Y}_n\}_{n \in \mathbb{N}}$ converges in distribution to an $\mathbb{R}^d$-valued random variable $\boldsymbol{Y}$, in short $\boldsymbol{Y}_n \xrightarrow{d} \boldsymbol{Y}$. If $\boldsymbol{X}_n$ and $\boldsymbol{Y}_n$ have the same distribution for all $n \in \mathbb{N}$, then $\boldsymbol{X}$ and $\boldsymbol{Y}$ have the same distribution.*

*Proof.* By Levy's continuity theorem [Durrett, 2019, Theorem 3.3.17],

$$\lim_{n \to \infty} \mathbb{E}[e^{i\boldsymbol{t} \cdot \boldsymbol{X}_n}] = \mathbb{E}[e^{i\boldsymbol{t} \cdot \boldsymbol{X}}]$$

and

$$\lim_{n \to \infty} \mathbb{E}[e^{i\boldsymbol{t} \cdot \boldsymbol{Y}_n}] = \mathbb{E}[e^{i\boldsymbol{t} \cdot \boldsymbol{Y}}]$$

for all $\boldsymbol{t} \in \mathbb{R}^d$. Because $\boldsymbol{X}_n$ and $\boldsymbol{Y}_n$ have the same distribution for all $n \in \mathbb{N}_{\geq 1}$,

$$\mathbb{E}[e^{i\boldsymbol{t} \cdot \boldsymbol{X}_n}] = \mathbb{E}[e^{i\boldsymbol{t} \cdot \boldsymbol{Y}_n}]$$

for all $\boldsymbol{t} \in \mathbb{R}^d$ and for all $n \in \mathbb{N}_{\geq 1}$. Therefore,

$$\mathbb{E}[e^{i\boldsymbol{t} \cdot \boldsymbol{X}}] = \mathbb{E}[e^{i\boldsymbol{t} \cdot \boldsymbol{Y}}]$$

for all $\boldsymbol{t} \in \mathbb{R}^d$.

Thus, $\boldsymbol{X}$ and $\boldsymbol{Y}$ have the same characteristic function and hence the same distribution [Durrett, 2019, Theorem 3.3.11]. $\qquad \square$

# D NUMERICAL ILLUSTRATION

We now consider a small numerical illustration of the violation of Assumption (A3).

**Example 3** (Numerical Illustration of the violation of Assumption (A3)). For different sample sizes and different $\phi$'s, we simulate $X_0^1$ from the given solution in equation (3): For that, we first generate the noise variables an then approximate the infinite sum with a finite sum consisting of the first 50 summands. We then estimate the covariance between $X_0^1$ and $X_1^2 = \epsilon_1^2$; the theoretical covariance between the two is $-\phi^{-2}$. For each combination of sample size and $\phi$, we do 100 different runs and calculate the mean and standard error of the estimated covariances.

We have implemented this example in R (version 3.6.3) [R Core Team, 2020] and provide the code in the attached supplementary material.

| $\phi$ | sample size | est_cov_mean | est_cov_se |
|---|---|---|---|
| 2 | $10^2$ | $-0.25$ | 0.01 |
| 2 | $10^3$ | $-0.25$ | 0.00 |
| 2 | $10^4$ | $-0.25$ | 0.00 |
| 2 | $10^5$ | $-0.25$ | 0.00 |
| 3 | $10^2$ | $-0.12$ | 0.01 |
| 3 | $10^3$ | $-0.11$ | 0.00 |
| 3 | $10^4$ | $-0.11$ | 0.00 |
| 3 | $10^5$ | $-0.11$ | 0.00 |
| 5 | $10^2$ | $-0.04$ | 0.00 |
| 5 | $10^3$ | $-0.04$ | 0.00 |
| 5 | $10^4$ | $-0.04$ | 0.00 |
| 5 | $10^5$ | $-0.04$ | 0.00 |

Table 1: Results for the numerical illustration of Example 3.

The numerical results correspond well to our theoretical considerations.