# OpenReview forum: "A Global Markov Property for Solutions of Stochastic Difference Equations and the corresponding Full Time Graphs"
_auai.org/UAI/2024/Conference — UAI 2024 poster_

### Official Review · Reviewer_JpY7 · 2024-03-14

**Q2-1 Originality-Novelty:** 3
**Q2-2 Correctness-Technical Quality:** 2
**Q2-5 Clarity Of Writing:** 2

**Q1 Summary And Contributions:**

This paper discusses discrete time series where the stochastic difference equations are in a form known from structural causal models.
They demonstrate then that a global Markov property for the full time graph of such a time series implied by the structural equations holds under standard assumptions. They find this result for the practically relevant case of a time series that started at time $-\infty$ such that equilibria can be considered.

**Q2-3 Extent To Which Claims Are Supported By Evidence:**

3: Good: the main claims are supported by convincing evidence (in the form of adequate experimental evaluation, proofs, (pseudo-)code, references, assumptions).

**Q2-4 Reproducibility:**

2: Fair: key resources (e.g. proofs, code, data) are unavailable but key details (e.g. proof sketches, experimental setup) are sufficiently well-described for an expert to confidently reproduce the main results.

**Q3 Main Strengths:**

The authors give formal proof that and under which assumption the Markov property can be inferred for full time graphs. Given the amount of time series data, extending this result from the i.i.d. case appears valuable. The assumptions they make seem reasonable and almost minimal.
The main Theorem is split into Lemmata 2 and 3. I like that those are presented in a way that the Theorem can be easily deduced given the Lemmata.

**Q4 Main Weakness:**

1. There is no justification for why the detour via the cutoff graph and m-separation is necessary for the argument.

2. It seems that a graph and its augmented version are linked trivially. I am not sure if it is worth providing always results for the two separately.
This leads to additional notation for the reader to cope with. Notation in time series is complex by nature. So introducing more burden than necessary is unwelcome.

3. The proof sketches, particularly for Lemma 3, which use mostly graphical arguments are hard to follow as there are no illustrations. In parts, I did not manage to understand it.

4. Instantaneous effects are just omitted directly. Acyclic instantaneous effects would still fit well in the SCM framework. So, this deserves more attention.

5. There are flaws in the definition of colliders in the supplementary material. As colliders then define, m- and d- separation, this has a major impact on the understanding of the results if people rely on these definitions.

**Q5 Detailed Comments To The Authors:**

- p1: ``Doing so hints at several pitfalls at the intersection of time series analysis and causal inference.'' Which pitfalls? Of course, violation of (A3) is an issue, but not a practically relevant one, based on my understanding of the world.

- p1: ``Moreover, we introduce a new projection procedure for these infinite graphs which might be of independent interest.'' The independent interest in this graph is not backed up.

- p2: ``so far only exists for vector autoregressive(VAR) processes with independent Gaussian errors''. On a quick skim, Thams et al. arrive at their result more directly while it is not evident to me where Gaussianity is necessary. Could you explain why their result is not expandable to the more general case?

- p2: Related to the point above. Do you have a comment on the correctness of the result in Dahlhaus and Eichler? Their important Theorem 3.3. refers to Andersson et al. 2001 (Alternative Markov properties for chain graphs.) for the proof. However, this reference does not appear to be suitable for infinite graphs.

- p2: One is directed to the SM for the graphical definitions. I think the definition of colliders on p11 has several flaws. There are four options for how $v$ could be a collider. None of them is a collider to the best of my knowledge. Two are equivalent up to symmetry, so arguably superfluous. Two are equivalent up to the named endpoint $w$.

- p2: Instead of a purely mathematical definition of $q_i$ that one only understands with the coming paragraph, it could be called the 'order' directly. Similarly, $PA_s^i$ could be given an intuitive name as a description.

- p3: ``We similarly define causal-parentship of a noise variable.'' This is a continuing confusion for me. It seems that $\epsilon_s^j$ is a parent of $X_t^i$ (if and) only if $i=j$, $t=s$ according to equation (1). If this understanding is true, why make it so complicated?

- p3: What does $i \perp [d]_1$ mean? $\epsilon_t^i$ is independent from all other $\epsilon_t^j$? This notation should be defined.

- p3: ``and a directed edge from $\epsilon_s^i$ to $X_t^i$'' Related to a previous comment. Should it be $\epsilon_t^i$ instead?

- p3: Remark 1 states that you do not consider instantaneous effects. However, allowing for acyclic instantaneous effects fits well within the structural causal model framework. Can you assess if/how your results would carry over to that case?

- p4: ``In Section 4, we will see that Assumption (A4) is satisfied for stable VAR processes.'' I find this statement a bit strong as you only give a reference from which the reader should deduce (A4).

- p6: ``Two auxiliary variables are in the same connected component if and only if there is a path between them in the respective cutoff graph just consisting of bidirected edges.'' Is this synonym to saying that the path includes auxiliary variables only?

- p7: I was not able to follow the paragraph from ``If $v$ is a collider...'' as well as Case 2 after a reasonable amount of reading time. Probably, this could be simplified a lot using graphical illustrations. I understand that this does not fit the page limit. But, maybe it is then not worth having this proof in the main text.

- p11: ``we say that there is a path between them if there is a vertex $v \in A$ and a vertex $b \in B$ such that there is a path between $v$ and $w$.'' Should it be $w \in B$?

**Q9 Complying With Reviewing Instructions:**

Yes

---

> ### Author Rebuttal · Authors · 2024-04-03
>
> We thank Reviewer JpY7 for their thorough review which will substantially increase the quality of our work. We also want to thank Reviewer gccG for acknowledging that our result is valuable, our assumptions are almost minimial, and that the Lemmatas are presented in a way that make the Theorem easily deducable. In the following, we will go through the points (the literature references, if not otherwise indicated, are as in our paper):
>
> **Instantaneous effects are just omitted directly. Acyclic instantaneous effects would still fit well in the SCM framework. So, this deserves more attention.**
>
> We want to note that our paper explicitly allows for acyclic instantaneous effects: Note that equation (1) has the term $PA_0$, which are instantaneous parents. Also note that the Assumption (A1) does not prohibit instantaneous effects; we just prohibit cycles. The sentences below that assumption just state that cycles are impossible in our setting if there are no instantaneous effects. However, due to an embarrassing typo in Remark 1, we indeed seem to exclude instantaneous effects. In Remark 1, it should be $s>t$ instead of $s\geq t$. We will fix this typo and write "directed cycle" instead of "cycle" in Assumption (A1) for clarification.
>
>
> **There is no justification for why the detour via the cutoff graph and m-separation is necessary for the argument.**
>
> We want to employ existing results for graphs where every vertex has finitely many ancestors. For that, we need to establish a correspondence between graphs that continue to $-\infty$ and graphs that start at some initial slice. This is the point where cutoff graphs come into play.
>
> Having said that, we do not claim that our proof idea is the only one, but we also want to note that we are not aware of another proof (for the global Markov property based only on the general assumptions that we are making.)
>
>
> **p1: "Moreover, we introduce a new projection procedure for these infinite graphs which might be of independent interest." The independent interest in this graph is not backed up.**
>
> As explained in the previous point, we establish a correspondence between graphs starting at $-\infty$ and graphs starting at some initial root vertices. The idea is then that one might use the latter graphs (starting at initial vertices) to reason about the graphs starting at $-\infty$. We imagine that this correspondence might be relevant in similar proofs or other application examples.
>
>
>
> **It seems that a graph and its augmented version are linked trivially. I am not sure if it is worth providing ...**
>
> Indeed, the augmented full time graph and full time graph are linked in a rather trivial way. However, because the proof for the full time graph relies on the result for the augmented full time graph (see proof of Lemma 2), we wanted to include these results to make the proof ideas available in the main part of the paper. Nevertheless, we understand that the inclusion of the augmented object in the main paper complicates reading, which is why we will now mainly move the points on the augmented full time graph to the Supplementary Material.
>
>
> **p2: "so far only exists for vector autoregressive(VAR) processes with independent Gaussian errors". On a quick skim, Thams et al. arrive at their result more directly while it is not evident to me where Gaussianity is necessary. Could you explain why their result is not expandable to the more general case?**
>
> Thams et al. have uploaded an updated version on March 15 2024, so while our paper was already in review. In the new version, the assumption of Gaussianity for that proof has been dropped. Now, only finite second moments are required, similarly to what we require for VAR processes. We will update the corresponding sentences in our paper.
>
> Having said that, our assumptions are still more general than the ones of Thams et al. Instead of assuming a particular stochastic process, we assume certain properties of stochastic processes that are quite general. We also think that cutting-off infinite graphs is also rather intuitive as this procedure builds a clear connection between two types of graphs that are commonly interchanged in practice.
>
> **Regarding the Dahlhaus and Eichler paper**: Indeed, the AMP paper from Andersson et al. 2001 assumes finitely many vertices. In our opinion, using the AMP property straight out of the box requires further justification that is not provided by Dahlhaus and Eichler.
>
> **Collider definition in the SM:**
> Indeed, the definition of colliders contains embarassing typos. We will fix that.
>
> We also thank the reviewer for the other pointed out typos which we will fix.
>
> **We will also make the proof of Lemma 3 more accessible using graphical illustratations.**
>
> **What does $i \perp [d]_1$ mean?
> is independent from all other
> ? This notation should be defined.** Another typo, it should be $i\in [d]_1$. Thanks for pointing it out.

---

### Official Review · Reviewer_gccG · 2024-03-20

**Q2-1 Originality-Novelty:** 2
**Q2-2 Correctness-Technical Quality:** 3
**Q2-5 Clarity Of Writing:** 1

**Q1 Summary And Contributions:**

Suppose we have an infinite stationary time series where the value of a variable at time t depends on some variable values within a finite time interval before and a noise term that is independent of all other variables. This time series can be represented by an infinite DAG. If we "cutoff" the time series and consider some chunk from time t forwards, some variables in the beginning of the chunk be have unobserved confounders due to us not observing the whole time series. This paper shows that, if variable sets S_1 and S_2 are d-separated by S_3 in the infinite graph then they are m-separated in the cutoff graph.

**Q2-3 Extent To Which Claims Are Supported By Evidence:**

3: Good: the main claims are supported by convincing evidence (in the form of adequate experimental evaluation, proofs, (pseudo-)code, references, assumptions).

**Q2-4 Reproducibility:**

3: Good: key resources (e.g. proofs, code, data) are available and key details (e.g. proofs, experimental setup) are sufficiently well-described for competent researchers to confidently reproduce the main results.

**Q3 Main Strengths:**

The paper bridges gaps between stochastic difference equations and probabilistic graphical models.

**Q4 Main Weakness:**

The paper was difficult the read. There was a mix of stochastic difference equation terminology and graphical model terminology and the connections between them were unclear.

One big problem is that the paper studies properties of solutions of stochastic difference equations. However, it is never defined what kind of an object is a "solution of stochastic difference equations".

In the standard language of graphical models, if two nodes of are d-separated, then the corresponding random variables are conditionally independent. If a "solution of stochastic difference equations" is not a random variable then one should precisely define it. (And if they are the same then it would be simpler to use the term "variable" instead.)

Due to the above confusion, the whole motivation of this work was unclear.

My understanding about solutions of stochastic difference equations is that you want to express X_t at time t as a function of the initial state and the epsilons. In this setup, the time series starting from minus infinity can be a problem. However, I do not see how this would be a problem from the d-separation point of view. Essentially, if some variables in the "past" are going to be d-separated from the variables in the "future" given some variables in the "present" then the "past" variables and "future" variables are conditionally independent of the "present" variables. So if we want to solve the system for "future" variables and have observed "present" values then we do not need to care whether the time series is infinite or not.

**Q5 Detailed Comments To The Authors:**

While talking about causality is fashionable these days, the paper does not do anything inherently causal so it is more natural to talk about Bayesian networks instead of causal models (SCMs are a special case of Bayesian networks).

Specifically, the paper is closely related to Dynamic Bayesian networks that are designed for temporal data. See, for example,

Uffe Kjærulff: dHugin: a computational system for dynamic time-sliced Bayesian networks. International Journal of Forecasting 11 (1995) 89-111


We can represent to stochastic difference equations in the language of Bayesian networks.

A Bayesian network consists of a structure which is a DAG G and local conditional distributions P(X | PA_X) where are the parents of X in G. Assuming that noises are independent (A0), equation (1) holds and the connections within a time steps do not create any cycles, we get a Bayesian network whose structure is is the augmented full time graph. We can just specify the conditional distribution for X as follows:

P(X = x | PA_X, epsilon_X) = 1 if x = f(PA_X, epsilon_X), 0 otherwise

Then, we have some marginal distribution P(epsilon_X) for each epsilon_X.

Once we have enough assumptions to represent the whole distribution as a Bayesian network, we have the known results for Bayesian networks at disposal.

If we assume stationarity, then we can represent the above distribution using a dynamic Bayesian network. But the Markov properties derived from a Bayesian network do not require us to assume this.

Furthermore, if the distribution factorises according to the augmented full time graph then assumption A3 follows automatically (because X_s<=t and epsilon_s>t are d-separated in the augmented full time graph given an empty set).

And a Bayesian network modeller would call the example of violating A4 as "cheating". As Z_1 and Z_2 are random variables, they should be included in our model. So Z_1 = f (PA_Z_1, epsilon_Z_1) and Z_2 = f (PA_Z_2, epsilon_Z_2). And if they are dependent then there has to be some contemporaneous arcs between them.


Observing that the whole system factorises according to a Bayesian network makes also proofs simpler. For example, Lemma 1 follows directly from basic properties Bayesian networks.

Proof: As the distribution factorises according a Bayesian network, d-separation in the graph implies conditional independence. Thus, if S_1 and S_2 are dependent then they cannot be d-separated by an empty set. Thus, there must exist at least one collider-free path between S_1 and S_2. The path to be collider-free, it must be either a directed path from S_1 to S_2, a directed path from S_2 to S_1 or a path with an internal node a (common ancestor) such that there is a directed path from a to both S_1 and S_2.

Note that the above proof is more general than in the paper. It does not assume stationarity.


"... Markov property for time series ... thus far exists only for ..." Markov property is a property of a model and either exists or does not exist regardless of whether we can prove its existence.

"Moreover, the solution might behave quite differently and carry quite different causal interpretations" What does this mean?

**Q9 Complying With Reviewing Instructions:**

Yes

---

> ### Author Rebuttal · Authors · 2024-04-03
>
> We thank Reviewer gccG for their review and for pointing us to an interesting related paper. However, we respectfully disagree with the criticism by the reviewer. In particular, as we explain in the following, we think that the proof idea that Reviewer gccG suggests does not work (the literature references, if not otherwise indicated, are as in our paper):
>
> Dynamic Bayesian Networks (DBNs) are build on a theory that relies on *finitely* many vertices, or at least on an initial time slice. It is highly nontrivial to study models that start at $-\infty$ by using models that start at initial time slices (the cutoff graphs in our proof and in particular Lemma 1 offer a connection here). And our main motivation is to study models starting at $-\infty$.
> Moreover, note that also the paper referenced by Reviewer gccG explains in Section 2.4 that DBNs *are not* independence graphs if the first time slice is not represented explicitly in the model, so DBNs do not accurately reflect the conditional independencies of the corresponding distribution.
>
> In addition, the proof idea of Reviewer gccG relies on the fact that joint distributions factorize according the (augmented) full time graph. First of all, it is not entirely clear what is meant by factorizing joint distributions for *infinitely* many variables; one would probably mean that every finite yet arbitrary large set of random variables factorize. However, without any further assumptions, this factorization does not always hold: For example, consider the example for a violation of Assumption (A3): As explained in this example, $X^1_{t} $ and $X^2_{t+1}$ are dependent given the empty set, so the joint distribution $P_{X^1_{t}X^2_{t+1}}$ does not equal the product distribution $P_{X^1_{t}}\otimes P_{X^2_{t+1}}$. When naively cutting off the graph by deleting every vertex and edge before time point $t$, neither $X^1_{t}$ nor $X^2_{t+1}$ have parents, so factorization according to the graph proclaims that $P_{X^1_{t}X^2_{t+1}}=P_{X^1_{t}}\otimes P_{X^2_{t+1}}$, contradiction. Note that by including more variables, this problem still occurs because the dependence structure over the initial time slices is not adequately represented in these naively cutted off graphs. One might be tempted to add bidirected edges now, and cut-off the graph differently, however, this then results in similar proofs that we actually discuss. If you so want, then our paper establishes this correspondence between starting at $-\infty$ and starting at some initial vertex (especially Lemma 1 establishes this correspondence).
>
> **One big problem is that the paper studies properties of solutions of stochastic difference equations. However, it is never defined what kind of an object ....**
>
> We kindly refer the reviewer to the paragraph above equation (1) where we write: " ... we assume that {$X_t$}$\_{t\in\mathbb{Z}}$ and {$\epsilon_t$}$\_{t\in\mathbb{Z}}$ solve the stochastic difference equation ...".
> This sentence means that our solutions of the stochastic difference equations are stochastic processes {$X_t$}$\_{t\in\mathbb{Z}}$ and {$\epsilon_t$}$\_{t\in\mathbb{Z}}$, or if you so want, the stochastic process {$(X_t, \epsilon_t)$}$\_{t\in\mathbb{Z}}$ (for us, a stochastic process is a collection of random variables). The nodes/vertices then refer to the stochastic process at that particular time point, so for example, $X^1_{-10}, X^2_1, X^2_3$ etc. (Note that this is in contrast to summary graphs where an entire component process {$X^i_t$}$\_{t\in\mathbb{Z}}$ is represented by *one* node). We will make that more clear in the paper.
>
> **And a Bayesian network modeller would call the example of violating A4 as "cheating"....**
>
> Indeed, it would be good to have $Z_1$ and $Z_2$ explicitly modelled and one could also change the underlying stochastic difference equations. However, our point here is that even if one thinks that one has every variable explicitly modelled, technical issues might arise when naively writing down stochastic difference equations for all the observed and modelled variables.
> That also relates to the point about contemporaneous arcs: Yes, one could model these pathological cases by including contemporaneous arcs, however, our point here is that these contemporaneous dependencies can occur *when not* modelling contemporaneous arcs.  Lastly, in a formal mathematical proof as we are providing it in our paper, we simply need to consider all cases---even if these cases appear to be rather contrived corner-cases.
>
> We also kindly disagree with the statement that SCMs are a special case of Bayesian networks. SCMs are also made up structural equations that functionally relate the random variables to each other (SCMs also allow to model strictly more than Bayesian networks, see Example 6.19 in in Peters et al. [2017]). Because the stoch. diff. equations resemble these structural equations (and prior work does so too, our paper resembles classical SCM work.

---

### Official Review · Reviewer_c7bp · 2024-03-22

**Q2-1 Originality-Novelty:** 2
**Q2-2 Correctness-Technical Quality:** 3
**Q2-5 Clarity Of Writing:** 2

**Q1 Summary And Contributions:**

The paper proves the general version of the global Markov property for time-series to apply it to the vector autoregressive processes with independent non-Gaussian noise.

**Q2-3 Extent To Which Claims Are Supported By Evidence:**

2: Fair: the main claims are somewhat supported by evidence (but the experimental evaluation may be weak, or does not match entirely with the claims, important baselines may be missing, proofs contain important ideas but lack rigor, algorithmic details are only discussed superficially, references are imprecise, assumptions are not sufficiently motivated or explicated, etc.).

**Q2-4 Reproducibility:**

3: Good: key resources (e.g. proofs, code, data) are available and key details (e.g. proofs, experimental setup) are sufficiently well-described for competent researchers to confidently reproduce the main results.

**Q3 Main Strengths:**

-The authors include some justification for the assumptions which the proof relies on.
-The authors proved a d-separation global Markov property for certain solutions of stochastic difference equations and the corresponding (augmented) full-time graphs when $t \in \mathcal{Z}$.
- The authors show which assumption is theoretically needed for the Global Markov property to hold with counterexamples.
- The proof details have been provided.

**Q4 Main Weakness:**

- The writing needs improvement. For example, the introduction copies the same text from the abstract: e.g. "unique observational distribution. A standard result states that in this acyclic case, the induced observational distribution satisfies a d-separation global Markov property relative to the induced graph..”
- The theoretical results look strong but are very limited based on strong assumptions e.g. assumption A2, as shown by the fact that the authors cannot come up with a counterexample where d-separation global Markov property when assumption 2 is being violated.
- I don’t see why it is not intuitive that when the assumption A3 is violated, the global Markov property won’t hold. If global Markov property were to hold, shouldn’t A3 always be satisfied based on the full-time graph structure?
- Lack of experiments to showcase how the theoretical results can be useful.

**Q5 Detailed Comments To The Authors:**

See above.

**Q9 Complying With Reviewing Instructions:**

Yes

---

> ### Author Rebuttal · Authors · 2024-04-03
>
> We thank Reviewer c7bp for their review and for acknowledging that the assumptions are well explained and illustrated using counterexamples. In the following, we will in detail address the points raised by the reviewer (the literature references, if not otherwise indicated, are as in our paper).
>
>
> **The writing needs improvement. For example, the introduction copies the same text from the abstract: e.g. "unique observational distribution. A standard result states that in this acyclic case, the induced observational distribution satisfies a d-separation global Markov property relative to the induced graph..”**
>
> Based on your points, and the other reviewer's points, we will try to make the paper more readable. In particular, we try to make the assumptions and proofs more accessible. We also include some further high-level explanations.
>
> **The theoretical results look strong but are very limited based on strong assumptions e.g. assumption A2, as shown by the fact that the authors cannot come up with a counterexample where d-separation global Markov property when assumption 2 is being violated.**
>
> While we in principal agree that Assumption (A2) is restrictive, we also want to underline that this assumption is very common in the classical (non-linear)-time-series literature, and in other works that link causal inference and time series, see for example Malinsky and Spirtes [2018]. In particular, note that the previous paper on the global Markov property from Thams et al. [2021] also implicitly assumes (A2) by assuming a stable VAR process where the noise has finite second moments (see Section 4 in our paper for this).
>
>
> There is not so much work on non-stationary settings; the reason is that non-stationarity settings are difficult to handle. Furthermore, non-stationary settings are often reduced to stationary settings by either assuming piecewise stationarity or differencing time series until they are stationary. In both cases, we think that starting with the stationary case is a useful first step.
>
> We also want to underline that we are able to come up with counterexamples for Assumption (A2), however, in these cases, one of the other assumptions is also not satisfied. We will make that point more clear in the paper and include such examples.
>
> Regarding the other assumptions: Again note that all the other assumptions are implicitly made by the previous paper that contained a global Markov property (among other things) from Thams et al. [2021] by assuming a stable VAR process. Relatively to that paper we would thus argue that our assumptions are much weaker.
>
> **I don’t see why it is not intuitive that when the assumption A3 is violated, the global Markov property won’t hold. If global Markov property were to hold, shouldn’t A3 always be satisfied based on the full-time graph structure?**
>
> You are right: If the global Markov property (for the augmented full time graph) holds, then Assumption (A3) is satisfied. However, our logical reasoning is the other way round: We detail assumptions from the time series and causal inference literature that allow us to prove a global Markov property. While Assumption (A3), an assumption that is standard in the time series literature (see the references in the main paper), is implied by the Markov property, it is initially not clear whether (A3) by itself already implies the Markov property. And, as the counterexamples show, Assumption (A3) alone is indeed not enough to prove a global Markov property. We will explain this better in the updated version of the paper.
>
> Please also note that the example for a violation of (A3) is rather long due to the fact that we wanted to show an example for the full time graph (instead of the  *augmented* full time graph), and the need to verify all the other assumptions (thus showing that (A3) is necessary to make relative to our other assumptions). If we would have just considered the augmented full time graph and just Assumption (A3), then, as you state, it is rather obvious that the global Markov property is violated.
>
> **Lack of experiments to showcase how the theoretical results can be useful.**
>
> We plan to illustrate our examples for the assumption violations and our main theorem by generating samples from corresponding stochastic processes and numerically checking conditional independencies/dependencies.
>
> We also want to remark that the primary use of our paper is to theoretically justify many time-series causal inference methods; we think that this is by itself useful (but we also understand that "usefulness" is a somewhat subjective notion).

---

### Official Review · Reviewer_9oT6 · 2024-03-22

**Q2-1 Originality-Novelty:** 2
**Q2-2 Correctness-Technical Quality:** 3
**Q2-5 Clarity Of Writing:** 4

**Q1 Summary And Contributions:**

The paper introduces a global Markov property for time series with index set $\mathbb{Z}$.
Here the link between time series and structural causal models is based on an interpretation of stochastic difference equations as structural assignments.
In particular, the paper generalizes previous work for vector autoregressive processes.

**Q2-3 Extent To Which Claims Are Supported By Evidence:**

3: Good: the main claims are supported by convincing evidence (in the form of adequate experimental evaluation, proofs, (pseudo-)code, references, assumptions).

**Q2-4 Reproducibility:**

4: Excellent: key resources (e.g. proofs, code, data) are available and key details (e.g. proof sketches, experimental setup) are comprehensively described for competent researchers to confidently and easily reproduce the main results.

**Q3 Main Strengths:**

The paper contributes to the interesting problem of linking time series analysis and causal inference using structural causal models.
The purpose of the paper is clear and the presentation is good.
Assumptions are discussed and illustrated in detail.

**Q4 Main Weakness:**

The assumptions seem very strict. For example, in (A0) the errors are required to be iid, instead of independent.
It remains unclear whether the assumptions can be weakened.

**Q5 Detailed Comments To The Authors:**

1. Can you discuss the "convention that a vertex is always an ancestor or a descendant of itself"? For instance, you could illustrate this with an example.
2. Could you weaken (A0) to simply independent noise variables?
3. Is there a notion of faithfulness in these models?

**Q9 Complying With Reviewing Instructions:**

Yes

---

> ### Author Rebuttal · Authors · 2024-04-03
>
> We thank Reviewer 9oT6 for their review and acknowledging that the problem contributes an interesting problem linking time series and causal inference and that the paper is well-written and the presentation is good. The main concern of Reviewer 9oT6 seems to be the restrictedness of the assumptions, which we discuss in the following (the literature references, if not otherwise indicated, are as in our paper):
>
> **Could you weaken (A0) to simply independent noise variables?**
>
> There are two parts in Assumption (A0) and we are not entirely sure to which part the reviewer refers. In the following, we go through the two options that might appear too restricted:
>
> - We assume that each noise variable $\epsilon_t$ has the *same distribution*: First, please note that this assumption does not mean that each component $\epsilon^i_t$ has the same distribution (i.e., $\epsilon^i_t$ and $\epsilon^j_t$ with $i \neq j$ can have a different distribution). Rather, the assumption means that the distribution of $\epsilon_t$ is independent of $t$ (i.e., $\epsilon^i_t$ and $\epsilon^i_{t^\prime}$ have the same distribution for all $i$, $t$, and $t^\prime$). Second, this assumption is, in fact, not required for the proofs; however, without this assumption one can get Assumption (A2), so stationarity of the $\{X_t\}_{t\in\mathbb{Z}}$, only for very pathological cases. Nevertheless, we plan to incorporate that distinction in the updated version of the paper to avoid this confusion.
>     (Please also note that the assumption that the noise variables $\epsilon_t$ having the same distribution for all times $t$ appears frequently in the time series - causality literature, see for example Thams et al. [2021] or Peters et al. [2013]).
> - We assume that the noise variables $\epsilon^i_t$ are *mutually* independent. Just assuming pairwise independence is not enough. Typically, in the SCM literature, "independent noise variables" means $P_{\epsilon_1,\ldots,\epsilon_n}=P_{\epsilon_1}\otimes \cdots \otimes P_{\epsilon_n}$, so the joint distribution factorizes. In this spirit, we could restate Assumption (A0) as $P_{\epsilon^{i_1}\_{t_1}\ldots \epsilon^{i_n}\_{t_n}}=P_{\epsilon^{i_1}\_{t_1}}\otimes \cdots \otimes P\_{\epsilon^{i_n}_{t_n}}$ for all finite sets of noise variables. This assumption then implies the independence assumption from (A0). We will make that implication also more clear.
>
>
> Regarding the other assumptions:
> We made assumptions that naturally appear either in the causal inference or time series literature and from there tried to prove a global Markov property:
>
> Assumption (A1) just enforces that there are no directed cycles in the graph; this assumption often occurs and we consider it rather standard.
>
> Assumption (A2) is indeed more restrictive, but, because non-stationarity is quite difficult to handle, still occurs often in the time series - causality literature (e.g., see Malinsky and Spirtes [2018] or Pamfil [2020]). In particular, note that the paper from Thams et al. [2021] which also contains a global Markov property (among other things) also implicitly makes this assumption by assuming a stable VAR process where the noise has finite second moments (see Section 4 in our paper for this).
>
> Moreover, results for non-stationary cases often rely on the stationary case, for example, by assuming piecewise stationarity. Thus, we think it is helpful to start with the stationary case but also consider it out of scope to discuss the non-stationary case further.
>
> Assumptions (A3) and (A4) are also quite standard in the time series literature as we have pointed out in the paper. Both assumptions are also implicitly assumed by Thams et al. [2021], see Section 4 in our paper for this.
>
> But in addition, we allow for general nonlinear equations in comparison to just linear equations as in Thams et al. [2021].
>
>
> **Can you discuss the "convention that a vertex is always an ancestor or a descendant of itself"? For instance, you could illustrate this with an example.**
>
> Ancestors of a vertex contain all parent-vertices, all grandparents, great-grandparents etc. The convention "that a vertex is always an ancestor" means that the particular vertex itself is included in this ancestor-set. Same for descendants. That is, for all vertices $X^i_t$, it holds that $X^i_t$ is both an ancestor and a descendant of itself (same for $\epsilon^i_t)$.
>
> In the main paper, we will use the additional 2 allowed pages to more clearly introduce what ancestors, descendants etc. are. There, we will also point out out what we mean by this convention by an example.
>
>
> **Is there a notion of faithfulness in these models?:**
> We suspect that there is such a notion. However, we think that it is far from trivial to study faithfulness for these models. Therefore, we have so far not discussed the topic of faithfulness and rather leave it to future work. We will put this point in our Outlook-Section.

---

### Official Review · Reviewer_T1hg · 2024-03-22

**Q2-1 Originality-Novelty:** 2
**Q2-2 Correctness-Technical Quality:** 2
**Q2-5 Clarity Of Writing:** 2

**Q1 Summary And Contributions:**

This paper proposes a new global Markov property suitable for handling solutions of stochastic difference equations in time series analysis. The article first introduces the application of structural causal models (SCMs) in causal inference and points out the technical challenges that exist when dealing with time series “start” at $-\infty$. To overcome these challenges, the authors prove a more general Markov property that applies to time series and corresponding full time graphs. The article discusses in detail the assumptions required for this property and examines violations of these assumptions, revealing potential problems at the intersection of time series analysis and causal inference.

**Q2-3 Extent To Which Claims Are Supported By Evidence:**

3: Good: the main claims are supported by convincing evidence (in the form of adequate experimental evaluation, proofs, (pseudo-)code, references, assumptions).

**Q2-4 Reproducibility:**

2: Fair: key resources (e.g. proofs, code, data) are unavailable but key details (e.g. proof sketches, experimental setup) are sufficiently well-described for an expert to confidently reproduce the main results.

**Q3 Main Strengths:**

This paper studies a global Markov property on graphs, which is very interesting, and studying a global Markov property on time series is very challenging.

**Q4 Main Weakness:**

I think overall the authors did interesting research, but I have some concerns listed below.

If the authors had provided experimental verification of the correctness of the theory, it would have made the article more complete.

If the author explains the mathematical symbols used in the text, such as $\mathbb{Z}$, $\mathbb{N}$, etc., it will make it easier for the reader to follow.

**Q5 Detailed Comments To The Authors:**

Regarding the \textbf{(A0)}:``$\{\epsilon _{t}\}_{t\in\mathbb{Z}}$ is an i.i.d. process with independent components", what would happen if $\epsilon_{t}$ was not independently and identically distributed? Then would Theorem 1 not be true?

Regarding Theorem 1: if all variables $X^{i}_{t}$ have no $t$, would Theorem 1 be equal to the standard global Markov property?

**Q9 Complying With Reviewing Instructions:**

Yes

---

> ### Author Rebuttal · Authors · 2024-04-03
>
> We kindly thank Reviewer T1hg for their review and for acknowledging that proving a global Markov property for time series is interesting yet very challenging.
> In the following, we will in detail address the points that Reviewer T1hg has made (the literature references, if not otherwise indicated, are as in our paper):
>
>
> **If the author explains the mathematical symbols used in the text, such as
> $\mathbb{Z}$, $\mathbb{N}$, etc., it will make it easier for the reader to follow.**
>
> We agree with Reviewer T1hg that extending our notation-section makes the paper clearer. In the updated version, we will make use of the two additional allowed pages to provide a more detailed introduction to the notation. Regarding the particular symbols mentioned by the reviewer: $\mathbb{Z}$ and $\mathbb{N}$ stand for the whole and natural numbers, respectively. We will also make it more clear by either writing $\mathbb{N}\_{\geq 1}$ or $\mathbb{N}_{\geq 0}$ to indicate whether we mean the natural numbers without $0$ or with $0$.
>
> **Regarding the \textbf{(A0)}: If $\{\epsilon_{t}\}_{t\in\mathbb{Z}}$ was not independently and identically distributed? Then would Theorem 1 not be true?**
>
> Actually, our proof does not explicitly make use of the assumption that the $\epsilon_{t}$ are identically distributed. However, our proof does use the $X_t$'s being stationary, which is assumption (A2), and we do not see how (A2) could hold without the $\epsilon_{t}$ being identically distributed $t$. Therefore, we opted for assuming the $\epsilon_{t}$ to be identically distributed. While we imagine that Theorem 1 also holds for non-stationary cases, we have not proven this so far and consider it future work (also note that non-stationarity is often modelled by assuming piece-wise stationarity or differencing time series until one reaches stationarity, so starting with the stationary case is a useful first step).
> We will make that distinction clearer in the main paper.
>
> At the moment, we do require independence of the noise variables $\epsilon_{t}$ for the proof.
> This assumption also often occurs for SCMs, see for example Bongers et al [2021].
> We also think that further relaxations are possible here, but do not consider it in the current paper.
>
> **Regarding Theorem 1: if all variables
>  have no $t$, would Theorem 1 be equal to the standard global Markov property?**
>
> If one just drops the index $t$ and has infinitely many variables without a particular time index, then one would need to think about SCMs for infinitely many variables as well. Here again, the standard results do not necessarily apply because one could again be in a setting where vertices have infinitely many ancestors. So in this case, dropping the time-index does not necessarily reduce to the standard global Markov property. However, with strong enough assumptions, one should get the standard global Markov property.
>
>
>
> **If the authors had provided experimental verification of the correctness of the theory, it would have made the article more complete.**
>
> We plan to illustrate our examples on assumption violations and our main theorem by generating samples from corresponding stochastic processes and numerically checking conditional independencies/dependencies.

---

### Meta-Review · Area_Chair_ixtn · 2024-04-16

This paper closes an important gap by providing a rigorous mathematical treatment of infinitely long causal DAGs. The author responses to reviewer’s concerns seemed rather convincing to me, e.g. the justification of the assumptions. I read parts of the paper myself and tend to be at least among the most positive reviewers. I also disagree with the statement that the paper does not do anything inherently causal.